# Model Reconciliation via Cost-Optimal Explanations in Probabilistic Logic Programming

**Yinxu Tang**
Washington University in St. Louis
`t.yinxu@wustl.edu`

**Stylianos Loukas Vasileiou**
New Mexico State University
`stelios@nmsu.edu`

**Vincent Derkinderen**
KU Leuven
`vincent.derkinderen@kuleuven.be`

**William Yeoh**
Washington University in St. Louis
`wyeoh@wustl.edu`

## Abstract

In human-AI interaction, effective communication relies on aligning the AI agent's model with the human user's mental model, a process known as model reconciliation. However, existing model reconciliation approaches predominantly assume deterministic models, overlooking the fact that human knowledge is often uncertain or probabilistic. To bridge this gap, we present a probabilistic model reconciliation framework that resolves inconsistencies in MPE outcome probabilities between an agent's and a user's models. Our approach is built on probabilistic logic programming (PLP) using ProbLog, where explanations are generated as cost-optimal model updates that reconcile these probabilistic differences. We develop two search algorithms – a generic baseline and an optimized version. The latter is guided by theoretical insights and further extended with greedy and weighted variants to enhance scalability and efficiency. Our approach is validated through a user study on explanation types and computational experiments showing that the optimized version consistently outperforms the generic baseline.

## 1 Introduction

In human-AI interaction, effective communication relies on aligning the AI agent's model with the human user's mental model, as mismatched understandings can make the agent's behavior seem inexplicable [1]. Model reconciliation offers a powerful explainable AI (XAI) approach by adjusting the human's model to align with the agent's understanding [2]. For instance, in planning, it explains why an agent's actions are valid in its model but not in the human's [2]. However, existing model reconciliation methods assume deterministic user beliefs, treating them as fixed or drawn from a set of distinct models [3, 4]. This overlooks the uncertainty and graded beliefs typical of human knowledge. In reality, humans often maintain probabilistic beliefs that reflect degrees of confidence rather than absolute truths. Ignoring this uncertainty can lead to unconvincing or misaligned explanations.

Probabilistic reasoning addresses uncertainty, capturing graded beliefs and uncertain outcomes. Inference methods like most probable explanation (MPE) and maximum a posteriori (MAP) are key: MPE finds the most likely scenario given evidence, while MAP identifies the most probable hypothesis [5]. These methods are widely applied, from Bayesian networks to probabilistic logic models. However, model reconciliation has yet to fully account for probabilistic beliefs. An agent may base decisions on an MPE outcome, while a human might disagree due to different probabilistic assumptions. Bridging this gap of reconciling probabilistic model differences between an agent and a human remains an open challenge.

Despite this gap, model reconciliation has been extensively explored in logic programming systems. In classical planning and answer set programming (ASP), explanations are generated by modifying

39th Conference on Neural Information Processing Systems (NeurIPS 2025).

logical rules to align beliefs [3, 6]. Meanwhile, probabilistic logic programming (PLP) frameworks like ProbLog [7] offer a powerful way to handle uncertainty, combining logical rules with probabilistic semantics. PLP supports diverse inference tasks, including MPE, MAP, and marginal probability computation, making it a versatile tool for uncertain reasoning. However, while PLP excels at probabilistic reasoning, its integration with model reconciliation for generating explanations under probabilistic beliefs remains unexplored.

In this paper, we introduce the first probabilistic model reconciliation framework within a PLP setting, leveraging ProbLog for its expressive power. Our approach allows an agent to reconcile differences between its probabilistic model and a human's model. The reconciliation is achieved by generating cost-optimal explanations that resolve inconsistencies in MPE outcome probabilities. Our key contributions are as follows:

- **Probabilistic Model Reconciliation:** We define model reconciliation under uncertainty, addressing inconsistencies in MPE outcome probabilities between an agent's and a human's ProbLog models.
- **Cost-Optimal Explanations:** We introduce a cost-based model where explanations are minimal updates that resolve probabilistic differences.
- **Algorithms and Scalability:** We develop two search algorithms – a generic baseline and an optimized version. The latter builds on theoretical insights to prune the search space and is further extended to greedy and weighted variants for improved scalability and efficiency.
- **Comprehensive Evaluation:** We validate our approach with a user study on explanation costs and computational tests showing the optimized search's superior performance.

## 2   Related Work

**Model Reconciliation Problems (MRPs).**   MRPs have been widely studied in domains like explainable planning and knowledge representation and reasoning (KR). In explainable planning [8], Chakraborti *et al.* defined it as aligning an agent's (planning) model with a human's by making minimal changes to the human model, using search methods like A* to balance explanation completeness and simplicity [2]. In KR, Vasileiou *et al.* introduced a general logic-based MRP framework for classical and hybrid planning problems [9], proposed a hitting-set algorithm to compute minimal sets of formulas that prove a target conclusion for problems beyond planning [4], and later extended this approach to generating personalized explanations [10]. Finally, Nguyen *et al.* reconciled ASP by identifying minimal rule additions and deletions such that the program yield the same target conclusion [3, 6]. Beyond deterministic models, Sreedharan *et al.* addressed uncertainty about the human's model by assuming that the human's model is located within a space of possible human models maintained by the agent [11], while Vasileiou *et al.* assumed that the human model is a probability distribution representing the agent's uncertainty of the actual human model [12].

These methods share a common limitation: they either assume user beliefs are deterministic or drawn from a fixed set of models, or do not allow for uncertain agent models. Consequently, these methods cannot reconcile differences when the agent's and human's models involve probability distributions over facts, rules, or outcomes. Our work addresses this gap through a joint probabilistic framework.

**Probabilistic Logic Programming (PLP).**   PLP integrates probabilistic reasoning with the expressive power of logic programming, enabling the specification of complex probabilistic models. This line of research started with Poole [13], who introduced the first PLP framework by extending the logic programming language Prolog [14], and with Sato [15], whose distribution semantics became the basis for several PLP systems, such as PRISM [16], ICL [17], ProbLog [7], and LPAD [18]. The notion of explanation has been explored by the PLP community [5, 19], where explanations are associated with possible worlds (i.e., truth-value assignments to all atoms in the language). The most prominent task there is that of the MPE, which consists of finding the world with the highest probability given some evidence [20]. However, a world does not show the chain of inferences of a given explanandum and it is not minimal by definition, since it usually includes a (possibly large) number of probabilistic facts whose truth value is irrelevant for the explanandum. An alternative approach is using the proof of an explanandum as an explanation [21], where a proof is a (minimal) partial world in which the query is true. In this case, one can easily ensure minimality, but even if the partial world contains no irrelevant facts, it is still not easy to determine the chain of inferences behind a given query. Finally, Renken *et al.* leveraged explanations in PLP as approximation techniques for more efficiently computing weighted model counting problems [22].

# 3 Background

We begin by reviewing the fundamental concepts of logic programming and its probabilistic extensions, with an emphasis on logical inference.

## 3.1 Logic Programming

An *atom* is an expression of the form $q(t_1, \ldots, t_n)$, where $q$ is a predicate of arity $n$, and each $t_i$ is a *term*. A term $t_i$ can be a *constant*, a *variable*, or a *functor* applied to other terms. A *literal* is either an atom or its negation $\neg q(t_1, \ldots, t_n)$. An expression is said to be *ground* if it contains no variables.

Syntactically, a *normal clause program*, or *logic program*, is a set of *rules*. A *rule* $r$ is an expression of the form $h :\!- b_1, \ldots, b_n$, where $h$ is an atom, referred to as the *head* of the rule, denoted by $\mathrm{head}(r) = h$. The *body* of the rule consists of a conjunction of literals $b_1, \ldots, b_n$, and is denoted by $\mathrm{body}(r) = \{b_1, \ldots, b_n\}$. The symbol ':−' represents logical implication ($\leftarrow$), and the comma ',' denotes the conjunction ($\wedge$). Thus, the rule states that $h$ holds whenever all literals in the body are satisfied. If $n = 0$, meaning the rule has an empty body, the rule is called a *fact*.

## 3.2 Probabilistic Logic Programming

**Syntax.** A ProbLog program $\mathcal{M}$ consists of a set of probabilistic facts $\mathcal{F}$ and a set of logic rules $\mathcal{R}$. Formally, the set of probabilistic facts can be written as $\mathcal{F} = \{f_1, f_2, \ldots, f_n\}$, where each $f_i$ is a ground fact. A *probabilistic fact*, written as $p_i :: f_i$, assigns a probability $p_i$ to the fact $f_i$, i.e., $P(f_i) = p_i$. Each fact is associated with a probability value.

Logic rules define deterministic dependencies between atoms (for simplicity, we assume that all atoms are ground). An atom that unifies with a probabilistic fact is called a probabilistic atom, whereas an atom that unifies with the head of a rule is referred to as a derived atom. We assume that the sets of probabilistic and derived atoms are disjoint.

**Semantics.** Each ground probabilistic fact $p_i :: f_i$ defines an *atomic choice*, in which $f_i$ is either included (with probability $p_i$) or excluded (with probability $1 - p_i$). A *total choice* is formed by making an atomic choice for each fact in $\mathcal{F}$, resulting in a subset $\mathcal{C} \subseteq \mathcal{F}$ of the selected facts. If there are $n$ probabilistic facts, the number of possible total choices is $2^n$. From those choices, we derive the remaining atoms by applying the logic rules.

The probability of a total choice $\mathcal{C}$ is computed by treating all atomic choices as independent events:

$$P(\mathcal{C}) = \prod_{f_i \in \mathcal{C}} p_i \cdot \prod_{f_i \in \mathcal{F} \backslash \mathcal{C}} (1 - p_i). \tag{1}$$

## 3.3 Inference

Given a ground atom $q$ (the *query*), the *relevant ground program* $\mathcal{M}(q)$ denotes the minimal subset of the grounded version of the original program $\mathcal{M}$ that is sufficient to derive $q$. Specifically, $\mathcal{M}(q)$ is obtained via backward reasoning from $q$, recursively identifying all probabilistic facts and rules necessary for its derivation. This process ensures that only the components relevant to the query are retained, thereby preserving correctness while enhancing the efficiency of probabilistic inference.

In model reconciliation, the *Most Probable Explanation* (MPE) inference is employed to identify the most probable set of assumptions that explain why a given query $q$ holds. This supports alignment between an agent and a human user by providing interpretable explanations. Formally, the MPE inference is defined as:

$$\mathrm{MPE}(q \mid \mathcal{M}) = \arg\max_{\mathcal{C}(q) \subseteq \mathcal{F}(q)} P(\mathcal{C}(q) \mid q), \tag{2}$$

where $\mathcal{F}(q)$ is the set of ground probabilistic facts in $\mathcal{M}(q)$, and $P(\mathcal{C}(q) \mid q)$ denotes the posterior probability of selecting the subset $\mathcal{C}(q)$ given that $q$ is observed to be true.

**Example 1.** *Consider the following ProbLog program $\mathcal{M}$ with query q = wet:*

|  |  |
|---|---|
| $0.7 :: \mathtt{rain}.$ | $0.7 :: a.$ |
| $0.7 :: \mathtt{sprinkler}.$ | $0.7 :: b.$ |
| $0.3 :: \mathtt{cloudy}.$ | $0.3 :: c.$ |
| $\mathtt{wet} :\!- \mathtt{rain}.$ | $d :\!- a.$ |
| $\mathtt{wet} :\!- \mathtt{sprinkler}.$ | $d :\!- b.$ |

*where* `rain`, `sprinkler`, `cloudy`, *and* `wet` *are denoted by a, b, c, and d, respectively, for simplicity.*

***Ground Probabilistic Facts of*** $\mathcal{M}(q)$: $\mathcal{F}(q) = \{a, b\}$, *since $c$ is irrelevant to query $d$.*

***Effective Choices***: $\mathcal{C}(q) \in \{\emptyset, \{a\}, \{b\}, \{a, b\}\}$. *For* $\mathcal{C}(q) = \{a, b\}$, $P(\mathcal{C}(q)) = 0.7 \times 0.7 = 0.49$.

***MPE Inference***: $\text{MPE}(q \mid \mathcal{M}) = \{a, b\}$, $P(\text{MPE}(q \mid \mathcal{M})) = 0.49$ *and* $\text{MPE}(\neg q \mid \mathcal{M}) = \emptyset$, $P(\text{MPE}(\neg q \mid \mathcal{M})) = 0.09$.

## 4   Probabilistic Model Reconciliation

We now present our framework for generating explanations in PLP. Intuitively, it enables us to resolve discrepancies between an agent's and a human's probabilistic models, caused by incomplete information, conflicting assumptions, or differing knowledge. Model reconciliation identifies and explains these differences, fostering a shared understanding between the agent and the human.

### 4.1   Problem Settings and Assumptions

We consider a setting where an agent and a human user each maintain their own ProbLog programs: the agent's model $\mathcal{M}_a$ and the human's model $\mathcal{M}_h$.

**Definition 1** (Model Inconsistency). *The agent model $\mathcal{M}_a$ and the human model $\mathcal{M}_h$ are said to be inconsistent with respect to a query $q$ if one of the following conditions holds:*

- ***Case 1****: $P(\text{MPE}(q|\mathcal{M}_a)) > P(\text{MPE}(\neg q|\mathcal{M}_a))$, but $P(\text{MPE}(q|\mathcal{M}_h)) < P(\text{MPE}(\neg q|\mathcal{M}_h))$*
- ***Case 2****: $P(\text{MPE}(q|\mathcal{M}_a)) < P(\text{MPE}(\neg q|\mathcal{M}_a))$, but $P(\text{MPE}(q|\mathcal{M}_h)) > P(\text{MPE}(\neg q|\mathcal{M}_h))$*

In both cases, the agent and the human assign opposite preferences to $q$ and $\neg q$, indicating a divergence in belief that motivates the need for model reconciliation.

### 4.2   Problem Formulation

To resolve the inconsistency between the agent and the human user, the agent must generate an explanation that allows the human to reconcile their model with that of the agent. To this end, we propose a logic-based formulation of model reconciliation within the ProbLog framework, referred to as a *P-MRP Explanation*. A P-MRP Explanation is formally defined as follows:

**Definition 2** (P-MRP Explanation). *Given that the agent model $\mathcal{M}_a$ and the human model $\mathcal{M}_h$ are inconsistent with respect to query $q$ (as defined in Definition 1), we define $\epsilon = \langle \epsilon^+, \epsilon^- \rangle$ as a P-MRP explanation for $q$ from $\mathcal{M}_a$ to $\mathcal{M}_h$ if and only if $\epsilon^+ \subseteq \mathcal{M}_a$, $\epsilon^- \subseteq \mathcal{M}_h$, and the updated human model $\mathcal{M}_h^* = (\mathcal{M}_h \cup \epsilon^+) \setminus \epsilon^-$ is both valid and consistent with $\mathcal{M}_a$ with respect to the query $q$.*

When the human model $\mathcal{M}_h$ is updated using a P-MRP explanation $\epsilon$, new formulae $\epsilon^+$ (including facts and rules) from $\mathcal{M}_a$ are added, and formulae $\epsilon^-$ from $\mathcal{M}_h$ are removed to ensure consistency.

To evaluate the quality of an explanation, we associate a cost with each candidate explanation $\epsilon = \langle \epsilon^+, \epsilon \rangle$, quantified by a cost function $\text{cost}(\epsilon)$ that models the cognitive effort required for the human to understand and incorporate the explanation. This function serves as the optimization objective and is defined as follows.

**Definition 3** (Explanation Cost). *Given an explanation $\epsilon = \langle \epsilon^+, \epsilon^- \rangle$, let $\epsilon_{fact}^+$ and $\epsilon_{fact}^-$ denote the sets of probabilistic facts, and $\epsilon_{rule}^+$ and $\epsilon_{rule}^-$ denote the sets of rules in $\epsilon^+$ and $\epsilon^-$, respectively. We consider the following types of modification:*[1]

- ***Change-probability*** *($c_p$): A cost $c_p$ is incurred for each fact $f_i \in \epsilon_{fact}^+ \cap \epsilon_{fact}^-$, representing a probability update.*
- ***Add-fact*** *($c_f^+$): A cost $c_f^+$ is incurred for each new fact $f_i \in \epsilon_{fact}^+ \setminus \epsilon_{fact}^-$.*
- ***Add-rule*** *($c_r^+$): A cost $c_r^+$ is incurred for each rule $r \in \epsilon_{rule}^+$ added to the model.*
- ***Delete-rule*** *($c_r^-$): A cost $c_r^-$ is incurred for each rule $r \in \epsilon_{rule}^-$ removed from the model.*

*The total explanation cost is given by:*

$$\text{cost}(\epsilon) = c_p \cdot |\epsilon_{fact}^+ \cap \epsilon_{fact}^-| + c_f^+ \cdot |\epsilon_{fact}^+ \setminus \epsilon_{fact}^-| + c_r^+ \cdot |\epsilon_{rule}^+| + c_r^- \cdot |\epsilon_{rule}^-|. \tag{3}$$

The task of explanation generation can be formulated as an optimization problem, defined as follows.

---

[1]We omit **delete-fact** since it is identical to **change-probability** that sets the probability of the fact to 0.

**Definition 4** (Optimal Explanation). *Let $\mathcal{E}_{valid}$ be the set of all valid P-MRP explanations $\epsilon$ as defined in Definition 2. Then, an optimal explanation is defined as: $\epsilon^* = \operatorname{argmin}_{\epsilon \in \mathcal{E}_{valid}} \operatorname{cost}(\epsilon)$.*

**Example 2.** *Continuing the scenario in Example 1, consider the following two models $\mathcal{M}_a$ and $\mathcal{M}_h$.*

$$
\mathcal{M}_a : \begin{array}{l} 0.7 :: a. \\ 0.7 :: b. \\ 0.3 :: c. \\ d : -a. \\ d : -b. \end{array} \qquad \mathcal{M}_h : \begin{array}{l} 0.3 :: a. \\ d : -a. \end{array}
$$

*Let the query be $d$. As shown in Example 1, we have $P(\operatorname{MPE}(d \mid \mathcal{M}_a)) > P(\operatorname{MPE}(\neg d \mid \mathcal{M}_a))$, but $P(\operatorname{MPE}(d \mid \mathcal{M}_h)) = P(\{a\}) = 0.3 < P(\operatorname{MPE}(\neg d \mid \mathcal{M}_h)) = P(\emptyset) = 0.7$.*

*As an illustration, consider the following valid P-MRP explanations for $d$ from $\mathcal{M}_a$ to $\mathcal{M}_h$: $\{\langle \{0.7 :: a.\}, \{0.3 :: a.\} \rangle, \langle \{0.7 :: b., d :\!- b.\}, \emptyset \rangle\}$. Given a modification cost of 1 per change, the optimal explanation is: $\epsilon^* = \langle \{0.7 :: a.\}, \{0.3 :: a.\} \rangle$, where $\operatorname{cost}(\epsilon^*) = 1$. This is a **change-probability** action, adjusting the probability of fact $a$ from $0.7$ to $0.3$.*

## 5 Search-Based Explanation Generation

Since computing a cost-optimal explanation under MPE semantics is NP-hard and lies in the $\Sigma_2^P$ complexity class [23], we propose two search-based algorithms to solve the optimization problem in Definition 4. The first is a generic search algorithm that exhaustively explores all explanations to find a cost-optimal one. To improve efficiency, we introduce an optimized search algorithm with pruning and cost-guided strategies.

Both algorithms construct explanations by incrementally modifying the human model. Actions are chosen from a two-level space: first, the type of modification (e.g., adding a fact or rule); second, the specific element to modify.

To formalize this process, we define the key notations of the agent and human models. Let $q$ be a query, and let $\mathcal{M}_a(q)$ and $\mathcal{M}_h(q)$ denote the relevant ground programs under the agent model $\mathcal{M}_a$ and the initial human model $\mathcal{M}_h$, respectively. The human model at timestep $t$ is denoted by $\mathcal{M}_{h,t}$, where $\mathcal{M}_{h,0} = \mathcal{M}_h$. Let $\mathcal{F}_a$ and $\mathcal{F}_{h,t}$ denote the sets of ground probabilistic facts in the agent model and the human model at timestep $t$. Similarly, let $\mathcal{F}_a(q)$ and $\mathcal{F}_{h,t}(q)$ represent the ground probabilistic facts appearing in the relevant programs $\mathcal{M}_a(q)$ and $\mathcal{M}_{h,t}(q)$, and let $\mathcal{R}_a(q)$ and $\mathcal{R}_{h,t}(q)$ denote the corresponding sets of rules. For any ground fact $f$, we use $P_a(f)$ and $P_{h,t}(f)$ to denote its probability in the agent and human models, respectively.

### 5.1 Generic Search Algorithm

The *generic search* algorithm exhaustively explores the explanation space without pruning. The first-level action space consists of four types of model modification operations, defined as:

$$
\mathcal{A}_{\text{type}} = \{\text{change-probability}, \text{add-fact}, \text{add-rule}, \text{delete-rule}\}. \tag{4}
$$

Given a selected action type $a_t \in \mathcal{A}_{\text{type}}$ at timestep $t$, the second-level action space specifies the candidate elements applicable under $a_t$:

- If $a_t = \text{change-probability}$, then the candidate space $A_t^c$ contains shared facts with different probabilities: $\mathcal{A}_t^c = \{f \mid f \in \mathcal{F}_a \cap \mathcal{F}_{h,t}(q), \ P_a(f) \neq P_{h,t}(f)\}$.
- If $a_t = \text{add-fact}$, then the candidate space is $\mathcal{A}_t^a = \mathcal{F}_a(q) \setminus \mathcal{F}_{h,t}$, representing facts available in the agent model but absent from the human model.
- If $a_t = \text{add-rule}$, then the candidate space is $\mathcal{A}_t^{r,+} = \{r \mid r \in \mathcal{R}_a(q) \setminus \mathcal{R}_{h,t}(q), \operatorname{body}(r) \subseteq \mathcal{F}_{h,t}\}$ containing rules that can be added to the human model.
- If $a_t = \text{delete-rule}$, then the candidate space is $\mathcal{A}_t^{r,-} = \mathcal{R}_{h,t}(q)$, consisting of rules in the human model that can be removed. Note that we do not consider deleting rules that were previously added.

At each timestep $t$, the explanation is represented as $\epsilon_t = \langle \epsilon_t^+, \epsilon_t^- \rangle$, where $\epsilon_0^+ = \epsilon_0^- = \emptyset$. The explanation is updated based on the selected action $a_t$ and element $e_t$ as follows:

$$
\langle \epsilon_t^+, \epsilon_t^- \rangle = \begin{cases} \langle \epsilon_{t-1}^+ \cup \{P_a(e_t) :: e_t.\}, \ \epsilon_{t-1}^- \cup \{P_{h,t}(e_t) :: e_t.\} \rangle & a_t = \text{change-probability}, \\ \langle \epsilon_{t-1}^+ \cup \{P_a(e_t) :: e_t.\}, \ \epsilon_{t-1}^- \rangle & a_t = \text{add-fact}, \\ \langle \epsilon_{t-1}^+ \cup \{e_t\}, \ \epsilon_{t-1}^- \rangle & a_t = \text{add-rule}, \\ \langle \epsilon_{t-1}^+, \ \epsilon_{t-1}^- \cup \{e_t\} \rangle & a_t = \text{delete-rule}. \end{cases} \tag{5}
$$

After each step, the human model is updated by $\mathcal{M}_{h,t} = (\mathcal{M}_h \cup \epsilon_t^+) \setminus \epsilon_t^-$, where $t \geq 0$.

The search process follows the A* algorithm, where the heuristic function $h_t^{\text{gen}}$ at each timestep $t$ is defined as the minimum cost among all possible action types:

$$h_t^{\text{gen}} = \min\{c_p, c_f^+, c_r^+, c_r^-\}, \tag{6}$$

where $c_p, c_f^+, c_r^+$, and $c_r^-$ denote the costs of change-probability, add-fact, add-rule, and delete-rule, respectively (see Definition 3). Further implementation details can be found in Appendix A.1.

To formally guarantee the correctness of this approach, we present the following validity theorem and the corresponding proof is provided in Appendix A.2.

**Theorem 1** (Validity Guarantee). *Given agent and human models $\mathcal{M}_a$ and $\mathcal{M}_h$ that are inconsistent with respect to a query $q$, the search procedure described above is guaranteed to find at least one valid explanation $\epsilon = \langle \epsilon^+, \epsilon^- \rangle$ such that the updated human model $\mathcal{M}_h^* = (\mathcal{M}_h \cup \epsilon^+) \setminus \epsilon^-$ is consistent with $\mathcal{M}_a$ regarding $q$.*

## 5.2 Optimized Search Algorithm

While the generic search algorithm is complete, it is often inefficient due to the large explanation space, where many actions are unnecessary for resolving the model inconsistency.

To enhance scalability, we introduce an *optimized search* algorithm that prunes irrelevant actions by focusing only on those needed to resolve the specific inconsistency. This approach identifies the minimal set of actions required, significantly reducing the search space without losing completeness.

To formalize this idea, we first introduce several definitions grounded in the ProbLog framework. These definitions provide the foundation for a pruning theorem and its proof, enabling precise reasoning about how model updates affect query outcomes.

**Definition 5** (DNF Representation of a Query). *Given a ProbLog program $\mathcal{M}$ and a query $q$, let $\mathcal{F}(q) = \{f_1, f_2, \ldots, f_n\}$ denote the set of probabilistic ground atoms in $\mathcal{M}(q)$. According to the semantics of ProbLog, the query $q$ can be represented as a disjunctive normal form (DNF) formula:*

$$q = \bigvee_{i=1}^m r_i, \quad \text{where} \quad r_i = \bigwedge_{j=1}^{k_i} a_i^j.$$

*Each clause $r_i$ corresponds to a derivation of $q$ and is expressed as a conjunction of literals. Each literal $a_i^j$ is either a ground atom or its negation, i.e., $a_i^j \in \mathcal{F}(q) \cup \{\neg f \mid f \in \mathcal{F}(q)\}$.*

This representation clarifies that satisfying any single conjunction $r_i$ is sufficient for $q$ to hold. Based on this structure, we now present the following theorem, which characterizes the relationship between the MPE probabilities and the DNF representation of the query.

**Theorem 2.** *Let $\mathcal{M}$ be a ProbLog program and $q$ a query with DNF representation $q = \bigvee_{i=1}^m r_i$, where $r_i = \bigwedge_{j=1}^{k_i} a_i^j$. Then:*

- **Case 1:** $P(\text{MPE}(q \mid \mathcal{M})) \geq P(\text{MPE}(\neg q \mid \mathcal{M})) \iff \exists i \in [m], \forall j \in [k_i], P(a_i^j) \geq 0.5$.
- **Case 2:** $P(\text{MPE}(q \mid \mathcal{M})) < P(\text{MPE}(\neg q \mid \mathcal{M})) \iff \forall i \in [m], \exists j \in [k_i], P(a_i^j) < 0.5$.

*Proof Sketch.* We present a proof sketch for **Case 1**, noting that the proof for **Case 2** proceeds analogously. Full details are provided in Appendix A.3.

($\Rightarrow$) *By contradiction:* Suppose that $q$ is more probable than $\neg q$ under MPE, yet each clause in its DNF contains a literal with a probability below $0.5$. Flipping any such literal would yield a more probable explanation favoring $\neg q$, contradicting the assumption that the MPE favors $q$.

($\Leftarrow$) If there exists a clause in the DNF of $q$ such that all its literals have probability at least $0.5$, then flipping them to true in the MPE of $\neg q$ results in an explanation that satisfies $q$ and is at least as probable. □

Based on Theorem 2, we can narrow the explanation search space by focusing on literals or clauses that are critical for switching the model's preference between $\neg q$ and $q$. This allows us to exclude actions irrelevant to belief change. We now refine the action space for each case in Definition 1.

**Case 1:** The agent prefers $q$ while the human prefers $\neg q$, i.e.,
$$P(\text{MPE}(q \mid \mathcal{M}_a)) > P(\text{MPE}(\neg q \mid \mathcal{M}_a)), \quad P(\text{MPE}(q \mid \mathcal{M}_h)) < P(\text{MPE}(\neg q \mid \mathcal{M}_h)).$$
According to Theorem 2, increasing $q$'s probability in $\mathcal{M}_h$ requires strengthening at least one DNF clause $r_i$ with all literals meeting $P(a_i^j) \geq 0.5$ in the updated model. This allows us to prune the first-level action space to:
$$\mathcal{A}_{\text{type}} = \{\text{change-probability, add-fact, add-rule}\}.$$
The **delete-rule** action is excluded because increasing $q$ only requires having *one* clause with all literals meeting the probability threshold, and deleting clauses does not help achieve this.

Given a selected action type $a_t \in \mathcal{A}_{\text{type}}$ at timestep $t$, the second-level action space defines the set of applicable candidate elements:

- If $a_t = $ change-probability, the candidate space is:
$$\mathcal{A}_t^c = \{f \mid f \in \mathcal{F}_a \cap \mathcal{F}_{h,t}(q), \ \text{sign}(P_a(f) - 0.5) \neq \text{sign}(P_{h,t}(f) - 0.5)\}, \tag{7}$$
  where $\text{sign}(\cdot)$ returns the sign of its input, capturing facts where the agent and human disagree on belief direction.
- If $a_t = $ add-fact, the candidate space remains the same as in the *generic search* algorithm.
- If $a_t = $ add-rule, we first define the current human belief set $\mathcal{B}_t$ based on the updated model $\mathcal{M}_{h,t}$:
$$\mathcal{B}_t = \{f \mid f \in \mathcal{F}_{h,t}, \ P_{h,t}(f) \geq 0.5\} \cup \{\neg f \mid f \in \mathcal{F}_{h,t}, \ P_{h,t}(f) \geq 0.5\}. \tag{8}$$
  That is, if the human assigns a probability strictly above 0.5 to $f$, we include $f$; if the probability is strictly below 0.5, we include its negation $\neg f$; and if $P_{h,t}(f) = 0.5$, both $f$ and $\neg f$ are included, reflecting a state of belief indifference.
  According to Theorem 2, addable rules $r$ must have all body literals supported by the current belief set, i.e., $\text{body}(r) \subseteq \mathcal{B}_t$. Additionally, to ensure $r$ can increase the probability, the body must include at least one literal $l$ where $P_{h,t}(l) \neq 0.5$. Therefore, the final candidate space is:
$$\mathcal{A}_t^{r,+} = \{r \mid r \in \mathcal{R}_a(q) \setminus \mathcal{R}_{h,t}(q), \ \text{body}(r) \subseteq \mathcal{B}_t, \ \exists l \in \text{body}(r) \ s.t. \ P_{h,t}(l) \neq 0.5\}. \tag{9}$$

**Case 2:** The agent prefers $\neg q$ while the human prefers $q$, i.e.,
$$P(\text{MPE}(q \mid \mathcal{M}_a)) < P(\text{MPE}(\neg q \mid \mathcal{M}_a)), \quad P(\text{MPE}(q \mid \mathcal{M}_h)) > P(\text{MPE}(\neg q \mid \mathcal{M}_h)).$$
According to Theorem 2, decreasing $q$'s probability in $\mathcal{M}_h$ requires weakening each DNF clause $r_i$ so that at least one of its literals satisfies $P(a_i^j) < 0.5$ in the updated model. This allows pruning the first-level action space as follows:
$$\mathcal{A}_{\text{type}} = \begin{cases} \{\text{change-probability, add-fact, add-rule}\}, & \text{if } \mathcal{R}_{h,t}(q) = \emptyset, \\ \{\text{change-probability, delete-rule}\}, & \text{if } \mathcal{R}_{h,t}(q) \neq \emptyset. \end{cases}$$
This distinction hinges on whether $\mathcal{R}_{h,t}(q)$ is empty. When $\mathcal{R}_{h,t}(q) = \emptyset$, new rules or facts can be introduced. Otherwise, only changing probabilities or deleting rules is effective, as each clause must include at least one literal with $P(a_i^j) < 0.5$.

Given a selected action type $a_t \in \mathcal{A}_{\text{type}}$ at timestep $t$, the second-level action space specifies the set of applicable candidate elements under $a_t$.

- If $a_t = $ change-probability, the candidate space is the same as in **Case 1**, as defined in Equation 7.
- If $a_t = $ add-fact, the candidate space is also identical to **Case 1**.
- If $a_t = $ add-rule, we define the opposing belief set as:
$$\mathcal{B}_t' = \{f \mid f \in \mathcal{F}_{h,t}, \ P_{h,t}(f) < 0.5\} \cup \{\neg f \mid f \in \mathcal{F}_{h,t}, \ P_{h,t}(f) > 0.5\}. \tag{10}$$
  We restrict addable rules $r$ such that at least one literal in the body of $r$ is supported by $\mathcal{B}_t'$. The candidate space is thus defined as:
$$\mathcal{A}_t^{r,+} = \{r \mid r \in \mathcal{R}_a(q) \setminus \mathcal{R}_{h,t}(q), \ \text{body}(r) \cap \mathcal{B}_t' \neq \emptyset\}.$$
- If $a_t = $ delete-rule, we allow the removal of rules whose bodies are not supported at all by the opposing belief set $\mathcal{B}_t'$. Formally,
$$\mathcal{A}_t^{r,-} = \{r \mid r \in \mathcal{R}_{h,t}(q), \ \text{body}(r) \cap \mathcal{B}_t' = \emptyset\}. \tag{11}$$

Similar to the generic version, we employ $A^*$ search to identify the cost-optimal explanation $\epsilon^*$, updating $\epsilon_t$ as in Equation 5. The refined heuristic function $h_t^{\text{opt}}$, derived from Theorem 2, is defined separately for two cases at each timestep $t$. It yields a more informative estimate that accelerates the search process compared with $h_t^{\text{gen}}$.

**Case 1:** When only *change-probability* actions are allowed, the heuristic estimates the minimal number of fact probability changes needed for each rule so that all literals in its body have probabilities of at least 0.5 in the updated model. For each rule $r \in \mathcal{R}_{h,t}(q)$, the set of modifiable literals is:

$$\mathcal{C}_t^r = \{\ell \mid \ell \in \mathrm{body}(r), P_{h,t}(\ell) < 0.5 \ \wedge \ (\ell \in \mathcal{A}_t^c \ \vee \ \neg\ell \in \mathcal{A}_t^c)\}, \tag{12}$$

where $\mathcal{A}_t^c$ is defined in Equation 7. A rule $r$ can be satisfied by probability adjustment alone only if every literal in its body is either already satisfied under $\mathcal{B}_t$ (Equation 8) or belongs to $\mathcal{C}_t^r$, i.e., $\mathrm{body}(r) \subseteq \mathcal{B}_t \cup \mathcal{C}_t^r$. In this case, the number of required modifications is $\delta_t^r = |\mathcal{C}_t^r|$; otherwise, we set $\delta_t^r = +\infty$, indicating that rule $r$ cannot be satisfied by probability adjustment alone.

When only *add-rule* actions are permitted, the heuristic evaluates whether introducing a new rule can restore consistency. If the set of addable rules $\mathcal{A}_t^{r,+}$ (defined in Equation 9) is non-empty, adding one rule suffices. Otherwise, at least one fact must be modified or added first.

The heuristic function is then defined as:

$$h_t^{\mathrm{opt}} = \begin{cases} \min(c_p \cdot \min_{r \in \mathcal{R}_{h,t}(q)} \delta_t^r, \ c_r^+), & \text{if } \mathcal{A}_t^{r,+} \neq \emptyset, \\ \min(c_p \cdot \min_{r \in \mathcal{R}_{h,t}(q)} \delta_t^r, \ c_r^+ + c_f^+, \ c_r^+ + c_p), & \text{if } \mathcal{A}_t^{r,+} = \emptyset. \end{cases} \tag{13}$$

**Case 2:** If $\mathcal{R}_{h,t}(q) = \emptyset$, a probability adjustment or at least one new rule must be introduced, implying that the heuristic function is defined as $h_t^{\mathrm{opt}} = \min(c_r^+, c_p)$.

Otherwise, the reasoning process depends on the available action types. When only *delete-rule* actions are permitted, we identify the rules that can be removed, namely those whose body literals all have probabilities not less than 0.5, corresponding to $\mathcal{A}_t^{r,-}$ as defined in Equation 11.

When only *change-probability* actions are allowed, the problem can be reformulated as a *set-cover* problem. We define the universe of unsatisfied rules as those not containing any literal $\ell$ that lies outside $\mathcal{A}_t^c$ (and its negation) with a probability less than 0.5:

$$\mathcal{U}_t = \mathcal{R}_{h,t}(q) \setminus \{r \mid r \in \mathcal{R}_{h,t}(q), \exists \ell \in \mathrm{body}(r), \ell \notin \mathcal{A}_t^c \wedge \neg\ell \notin \mathcal{A}_t^c \wedge P_{h,t}(\ell) < 0.5\}. \tag{14}$$

For each modifiable fact $f \in \mathcal{A}_t^c$, the cover subsets are defined as:

$$S_t^f = \{r \mid r \in \mathcal{U}_t, \exists \ell \in \{f, \neg f\} \ s.t. \ \ell \in \mathrm{body}(r) \wedge P_{h,t}(\ell) \geq 0.5 \wedge P_a(\ell) < 0.5\}. \tag{15}$$

The goal is to find the smallest collection of such subsets that covers all rules in $\mathcal{U}_t$:

$$\mathcal{S}_t^* = \arg\min_{\mathcal{S}' \subseteq \{S_t^f \mid f \in \mathcal{A}_t^c\}} |\mathcal{S}'| \quad s.t. \quad \bigcup_{S \in \mathcal{S}'} S = \mathcal{U}_t. \tag{16}$$

This formulation minimizes the number of change-probability actions needed so that each rule in $\mathcal{U}_t$ contains at least one literal with probability below 0.5. Since both $f$ and $\neg f$ may appear across rules, $\mathcal{S}_t^*$ may be empty, in which case at least one rule must be deleted to restore consistency.

If both *delete-rule* and *change-probability* actions are allowed, let $x$ ($x < |\mathcal{A}_t^{r,-}|$) denotes the number of deleted rules and $y$ the number of change-probability actions required to reach the optimal explanation from the current state. Removing $x$ rules from $\mathcal{U}_t$ can reduce the minimal cover size by at most $x$, which gives $|\mathcal{S}_t^*| - x \leq y \leq |\mathcal{S}_t^*|$. Consequently, the total cost from the current state to the optimal explanation satisfies $x \cdot c_r^- + y \cdot c_p \geq |\mathcal{S}_t^*| \cdot \min(c_r^-, c_p)$.

Based on the above analysis, the heuristic function is defined as:

$$h_t^{\mathrm{opt}} = \begin{cases} \min(|\mathcal{S}_t^*| \cdot \min(c_r^-, c_p), \ |\mathcal{A}_t^{r,-}| \cdot c_r^-), & \text{if } \mathcal{S}_t^* \neq \emptyset, \\ c_r^-, & \text{if } \mathcal{S}_t^* = \emptyset. \end{cases} \tag{17}$$

Solving the exact set-cover is NP-hard. To mitigate computational cost, the rule coverage can be restricted to those with the smallest body size: $\mathcal{U}_t = \{r \in \mathcal{U}_t \mid |\mathrm{body}(r)| = \min_{r' \in \mathcal{U}_t} |\mathrm{body}(r')|\}$. More details can be found in Appendix A.4.

To further improve efficiency, we introduce a greedy variant $h_t^{\mathrm{greedy}}$, which applies the standard greedy algorithm to the relaxed set-cover formulation. Building on *Weighted A\** properties, the resulting explanation cost remains bounded when different heuristics are employed, formalized below.

**Theorem 3** (Theoretical Guarantee). *Let $h_t^*$ denote the optimal heuristic value, $c^{\mathrm{opt}}$ the optimal explanation cost, and $w \geq 1$ the weight of heuristics used in Weighted A\*.*

- *Both $h_t^{\text{gen}}$ and $h_t^{\text{opt}}$ are admissible, i.e., $h_t^{\text{gen}} \leq h_t^*$ and $h_t^{\text{opt}} \leq h_t^*$. Consequently, Weighted $A^*$ employing $w \cdot h_t^{\text{gen}}$ or $w \cdot h_t^{\text{opt}}$ as the heuristic guarantees a total cost bounded by $w \cdot c^{\text{opt}}$.*
- *The greedy heuristic $h_t^{\text{greedy}}$ achieves a $(1 + \ln |\mathcal{U}_t|)$ approximation, i.e., $h_t^{\text{greedy}} \leq (1 + \ln |\mathcal{U}_t|) \cdot h_t^*$, where $\mathcal{U}_t$ is the universe of unsatisfied rules (see Equation 14). Hence, Weighted $A^*$ using $w \cdot h_t^{\text{greedy}}$ as the heuristic guarantees a total cost bounded by $w \cdot (1 + \ln(\max_t |\mathcal{U}_t|)) \cdot c^{\text{opt}}$, where $\max_t |\mathcal{U}_t| \leq |\mathcal{R}_{h,0}(q)|$ and $\mathcal{R}_{h,0}(q)$ is the set of rules in the initial human model relevant to query $q$.*

The detailed proof is provided in Appendix A.5.

## 6 Empirical Evaluations

### 6.1 Estimating Action Costs via Human-User Study

This study examines how an AI agent, *Blitzcrank*, explains its decisions in an intelligent warehouse, determining whether goods should be delivered. Participants compared explanation pairs and chose the one they felt best clarified the agent's reasoning. This setup represents one feasible way to evaluate explanation cost, though not the only possible approach.

**Data Collection.** We recruited 128 participants via Prolific [24], ensuring a diverse sample.[2] Participants were fluent English speakers and were compensated USD 2.00. After attention and coherence checks, data from 100 participants were retained for analysis.

**Cost Estimation.** To estimate the relative cognitive effort of different explanation types, we used the Bradley-Terry model [25] on the pairwise comparison data. Each action $a_i$ had a strength parameter $\beta_i$, with higher $\beta_i$ values indicating greater participant preference. The cost of an action was defined as the negative of its strength [26], $-\beta_i$, making more preferred actions correspond to lower costs. To ensure non-negative costs compatible with search algorithms (e.g., $A^*$), we exponentiated the negated strength values. The final cost of each action was defined $\text{cost}(a_i) = e^{-\beta_i}$. Based on this formulation, the estimated costs for the four explanation actions are shown in Table 1.

Table 1: Estimated Costs for Each Explanation Action.

| action $a_i$ | cost$(a_i)$ |
|---|---|
| change-probability | 0.9801 |
| add-fact | 0.8688 |
| add-rule | 1.0202 |
| delete-rule | 1.1511 |

### 6.2 Computational Results

This section evaluates the computational results of *Generic Search* and *Optimized Search* algorithms for model reconciliation. While the algorithms are general to any cost setting, the experiments use the costs listed in Table 1. All experiments were run on a MacBook Pro (M2, 16GB RAM).

**Experimental Setup.** Our experiments use two models: Agent and Human.

- Agent Model $\mathcal{M}_a$: Each $\mathcal{M}_a$ contains $|\mathcal{F}_a| = 10, 20, 100$, or $1000$ probabilistic facts and $|\mathcal{R}_a| = 5$, $10, 50$, or $500$ rules, respectively, all related to the same query. Facts have randomly assigned probabilities, and rules have bodies of 2-4 literals, generated based on cases in Definition 1.
- Human Model $\mathcal{M}_h$: Derived from each $\mathcal{M}_a$ at four complexity levels $l \in \{20\%, 40\%, 60\%, 80\%\}$, reflecting the percentage of probabilistic facts that differ. Each differing fact has a 1/3 chance of being: modified (probability flipped), removed, or replaced (new fact). Human model rules share the same heads as the agent model but are built using existing facts, with the rule count as: $|\mathcal{R}_h| = \lfloor |\mathcal{R}_a| \cdot (1 - 1/3 \cdot l) \rfloor$.

We generate 100 Agent-Human Model pairs for each configuration, totaling 1,600 pairs (4 Agent settings $\times$ 4 complexity levels $\times$ 100 repetitions) for each case in Definition 1.

**Evaluation Metrics.** All experiments are capped at 600 seconds per run.

- Average Time: The average runtime (in milliseconds) for runs completed within the time limit.
- Average Cost: The mean cost of achieved explanation for runs completed within the time limit.

**Results and Analysis.** Table 2 compares the two search algorithms under the cases defined in Definition 1. In **Case 1**, we report runtime for the generic ($h_t^{\text{gen}}$) and optimized ($h_t^{\text{opt}}$) heuristics, along

---

[2]Ethics approval was obtained from our university's IRB. The human-subject study, collected data, and implementation are released on `https://github.com/YODA-Lab/ProbLog-Model-Reconciliation`.

Table 2: Performance Comparison of Two Algorithms across Two Cases in Definition 1.

| $|\mathcal{F}_a|$ | $|\mathcal{R}_a|$ | $l$ | Case 1 | | | Case 2 | | | | | | | | |
| | | | Generic | Optimized | | Generic | Optimized | | | | | | | |
| | | | $h_t^{\text{gen}}$ (ms) | $h_t^{\text{opt}}$ (ms) | $c^{\text{opt}}$ | $h_t^{\text{gen}}$ (ms) | $h_t^{\text{opt}}$ (ms) | $c^{\text{opt}}$ | $h_t^{\text{greedy}}$ (ms) | $c^{\text{greedy}}$ | $2 \cdot h_t^{\text{opt}}$ (ms) | $c_2^{\text{opt}}$ | $2 \cdot h_t^{\text{greedy}}$ (ms) | $c_2^{\text{greedy}}$ |
| 10 | 5 | 20% | 28.7 | 34.0 | 1.46 | 36.1 | 33.9 | 1.64 | 34.9 | 1.64 | 33.2 | 1.64 | 35.9 | 1.64 |
| | | 40% | 29.7 | 34.9 | 1.67 | 36.2 | 34.7 | 1.47 | 34.9 | 1.47 | 33.9 | 1.47 | 35.9 | 1.47 |
| | | 60% | 35.8 | 33.9 | 1.95 | 30.8 | 33.9 | 1.37 | 35.8 | 1.37 | 33.8 | 1.37 | 35.7 | 1.37 |
| | | 80% | 33.6 | 35.8 | 2.14 | 28.1 | 34.5 | 1.28 | 36.2 | 1.28 | 33.8 | 1.28 | 36.9 | 1.28 |
| 20 | 10 | 20% | 1266.8 | 34.7 | 1.19 | 29758.7 (5 t/o)+ | 34.6 | 2.47 | 36.8 | 2.47 | 34.3 | 2.47 | 36.1 | 2.47 |
| | | 40% | 1846.7 | 35.4 | 1.22 | 19290.4 (1 t/o) | 35.2 | 2.08 | 36.0 | 2.08 | 34.4 | 2.08 | 35.0 | 2.08 |
| | | 60% | 3546.3 | 35.7 | 1.38 | 11657.0 (1 t/o) | 35.0 | 1.92 | 36.5 | 1.92 | 34.9 | 1.93 | 36.3 | 1.93 |
| | | 80% | 4406.2 | 34.6 | 1.30 | 3187.1 | 33.8 | 1.84 | 36.7 | 1.84 | 34.3 | 1.84 | 35.9 | 1.84 |
| 100 | 50 | 20% | –* | 37.0 | 0.99 | – | 40.0 | 8.98 | 38.0 | 8.98 | 37.7 | 8.99 | 35.57 | 8.99 |
| | | 40% | – | 37.7 | 0.98 | – | 228.1 | 7.93 | 53.9 | 7.93 | 139.0 | 7.93 | 40.7 | 7.93 |
| | | 60% | – | 39.0 | 0.98 | – | 3572.2 | 7.15 | 112.6 | 7.16 | 1980.9 | 7.15 | 43.0 | 7.16 |
| | | 80% | – | 39.2 | 0.98 | – | 8774.9 (1 t/o) | 5.48 | 136.6 | 5.48 | 3874.1 | 5.50 | 45.0 | 5.50 |
| 1000 | 500 | 20% | – | 176.5 | 0.98 | – | – | – | – | – | – | – | 889.1 | 80.40 |
| | | 40% | – | 235.6 | 0.98 | – | – | – | – | – | – | – | 3122.3 | 72.45 |
| | | 60% | – | 284.7 | 0.98 | – | – | – | – | – | – | – | 7554.6 | 63.34 |
| | | 80% | – | 331.1 | 0.98 | – | – | – | – | – | – | – | 18936.6 | 57.02 |

+ "t/o" indicates a timeout.
* "–" denotes that most runs timed out.

with the corresponding optimal costs $c^{\text{opt}}$. In **Case 2**, we further evaluate the greedy heuristic ($h_t^{\text{greedy}}$) and the weighted variants ($2 \cdot h_t^{\text{opt}}$ and $2 \cdot h_t^{\text{greedy}}$), reporting the resulting costs $c^{\text{greedy}}$, $c_2^{\text{opt}}$, and $c_2^{\text{greedy}}$.

- **Runtime Comparison.** The optimized heuristic $h_t^{\text{opt}}$ substantially reduces runtime compared to the generic $h_t^{\text{gen}}$ across all settings. For instance, in **Case 1**, runtime decreases by 99.2% (from 4406.2 ms to 34.6 ms) when $|\mathcal{F}_a| = 20$, $|\mathcal{R}_a| = 10$, and $l = 80\%$. In **Case 2**, the greedy heuristic $h_t^{\text{greedy}}$ further improves efficiency over $h_t^{\text{opt}}$. Moreover, the weighted variants $2 \cdot h_t^{\text{opt}}$ and $2 \cdot h_t^{\text{greedy}}$ offer additional runtime gains. For example, under $|\mathcal{F}_a| = 100$, $|\mathcal{R}_a| = 50$, and $l = 80\%$, $h_t^{\text{greedy}}$ achieves a 98.4% runtime reduction (from 8774.9 ms to 136.6 ms) compared with $h_t^{\text{opt}}$, while $2 \cdot h_t^{\text{opt}}$ achieves a 55.9% reduction over $h_t^{\text{opt}}$. Notably, $2 \cdot h_t^{\text{greedy}}$ can effectively produce valid solutions within the time limit, supported by the theoretical guarantee in Theorem 3.
- **Cost Comparison.** Focusing on **Case 2**, for $|\mathcal{F}_a| \in \{10, 20, 100\}$, the observed costs $c^{\text{greedy}}$, $c_2^{\text{opt}}$, and $c_2^{\text{greedy}}$ remain close to the optimal $c^{\text{opt}}$. Consistent with Theorem 3 (with $w = 2$), the empirical results satisfy the bounds $c^{\text{greedy}} \leq (1 + \ln|\mathcal{R}_h|) \cdot c^{\text{opt}}$, $c_2^{\text{opt}} \leq 2 \cdot c^{\text{opt}}$, and $c_2^{\text{greedy}} \leq 2 \cdot (1 + \ln|\mathcal{R}_h|) \cdot c^{\text{opt}}$. These results empirically corroborate our theoretical guarantees.

Further detailed analysis is provided in Appendix A.6. Overall, *Optimized Search* demonstrates superior scalability and robustness compared to *Generic Search*, consistently achieving lower average runtime. The greedy and weighted variants efficiently handle large-scale models while producing explanations that satisfy the theoretical guarantees established in Theorem 3. These findings underscore the effectiveness of theoretically guided search strategies for scalable model reconciliation.

## 7   Conclusion and Discussion

In this paper, we present a model reconciliation framework within probabilistic logic programming (PLP). Our approach formalizes reconciliation under uncertainty using ProbLog to represent an agent's and a human's probabilistic models, identifying and resolving inconsistencies in MPE outcome probabilities. We introduce a cost-based explanation model that quantifies the cognitive effort of model updates, enabling the generation of cost-optimal explanations that minimally adjust the human's model. To generate these explanations, we develop two search algorithms: a generic search algorithm and an optimized search algorithm guided by theoretical insights for pruning the search space. The optimized algorithm is further extended with greedy and weighted variants to improve scalability and runtime efficiency. We validate the framework through a user study examining how explanation types affect user understanding and through computational evaluations demonstrating that the optimized search consistently outperforms the generic method in both runtime and scalability.

Our framework enhances human-AI interaction by providing clear, cost-optimal explanations for AI decisions, thereby improving user understanding and trust across domains. By aligning the user's mental model with that of the agent, it enables more informed and transparent decision-making. Despite its strengths, the framework has limitations. As a logic-based approach, it has been evaluated mainly through computational experiments and a controlled user study, leaving its effectiveness in real-world settings to be further verified. In future work, we plan to incorporate techniques for learning user models from feedback to enable personalized and adaptive explanations [27].

## Acknowledgments and Disclosure of Funding

We thank all the reviewers for their insightful comments and suggestions, which significantly improved this paper. This research is partially supported by the National Science Foundation under Award No. 2232055. Additional support is provided by the Flemish Government (AI Research Program), the European Research Council (ERC) under the European Union's Horizon Europe research and innovation programme (Grant Agreement No. 101142702), and the iBOF/21/075 project. The views and conclusions expressed in this paper are those of the authors and do not necessarily reflect the official policies or positions of the sponsoring organizations, agencies, or the United States government.

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

# A Appendix

## A.1 Details of Generic Search Algorithm

This section presents the pseudocode of the generic search algorithm, together with an illustrative example of the action space used to construct candidate explanations.

---

**Algorithm 1** Generic Search Algorithm for Cost-Optimal Explanations

---

**Require:** Agent model $\mathcal{M}_a$, human model $\mathcal{M}_h$, query $q$
**Ensure:** Cost-optimal explanation $\epsilon^* = \langle \epsilon^+, \epsilon^- \rangle$
1: Initialize priority queue $\mathcal{Q} \leftarrow \{(\epsilon_0, g(\epsilon_0) = 0)\}$, where $\epsilon_0 = \langle \emptyset, \emptyset \rangle$
2: Initialize $g$-score map $\mathcal{G}[\epsilon_0] \leftarrow 0$ and closed set $\mathcal{V} \leftarrow \emptyset$
3: **while** $\mathcal{Q}$ is not empty **do**
4:     Pop $(\epsilon_t, g_t)$ with lowest $f_t = g_t + h_t^{\text{gen}}$
5:     **if** $g_t > \mathcal{G}[\epsilon_t]$ **then**
6:         **continue**
7:     **end if**
8:     **if** $\epsilon_t \in \mathcal{V}$ **then**
9:         **continue**
10:     **end if**
11:     Construct modified human model $\mathcal{M}_{h,t} \leftarrow (\mathcal{M}_h \cup \epsilon_t^+) \setminus \epsilon_t^-$
12:     **if** IsConsistent$(\mathcal{M}_{h,t}, \mathcal{M}_a, q)$ **then**
13:         **return** $\epsilon_t$
14:     **end if**
15:     Add $\epsilon_t$ to $\mathcal{V}$
16:     **for all** action types $a_t \in \mathcal{A}_{\text{type}}$ **do**
17:         Determine candidate set $\mathcal{A}_t^{a_t}$ according to Section 5.1
18:         **for all** elements $e_t \in \mathcal{A}_t^{a_t}$ **do**
19:             Generate successor $\epsilon_{t+1}$ according to Equation 5
20:             $g_{t+1} \leftarrow \text{cost}(\epsilon_{t+1})$
21:             **if** $\epsilon_{t+1} \notin \mathcal{G}$ **or** $g_{t+1} < \mathcal{G}[\epsilon_{t+1}]$ **then**
22:                 $\mathcal{G}[\epsilon_{t+1}] \leftarrow g_{t+1}$
23:                 Insert $(\epsilon_{t+1}, g_{t+1})$ into $\mathcal{Q}$ with priority $f_{t+1} = g_{t+1} + h_{t+1}^{\text{gen}}$
24:             **end if**
25:         **end for**
26:     **end for**
27: **end while**
28: **function** IsConsistent$(\mathcal{M}_{h,t}, \mathcal{M}_a, q)$
29:     **if** $P(\text{MPE}(q \mid \mathcal{M}_a)) > P(\text{MPE}(\neg q \mid \mathcal{M}_a))$ **then**
30:         **return** $P(\text{MPE}(q \mid \mathcal{M}_{h,t})) > P(\text{MPE}(\neg q \mid \mathcal{M}_{h,t}))$
31:     **else if** $P(\text{MPE}(q \mid \mathcal{M}_a)) < P(\text{MPE}(\neg q \mid \mathcal{M}_a))$ **then**
32:         **return** $P(\text{MPE}(q \mid \mathcal{M}_{h,t})) < P(\text{MPE}(\neg q \mid \mathcal{M}_{h,t}))$
33:     **end if**
34: **end function**

---

Algorithm 1 specifies a generic A* search framework for exploring the space of explanations. Starting from the empty explanation $\epsilon_0 = \langle \emptyset, \emptyset \rangle$, the algorithm incrementally applies actions to generate successor explanations and prioritizes them according to the evaluation function $f(\epsilon) = g(\epsilon) + h^{\text{gen}}(\epsilon)$. At each expansion step, the generated explanation is governed by a predefined set of action types and their corresponding candidate sets, which together define the action space of the search.

To clarify how this action space is instantiated in practice, we consider the following example:

$$\mathcal{M}_a : \begin{array}{l} 0.1 :: a. \\ 0.6 :: b. \\ 0.7 :: c. \\ 0.8 :: e. \\ d : -c, e. \\ d : -a, b. \\ d : -a, c. \end{array} \qquad \mathcal{M}_h : \begin{array}{l} 0.2 :: a. \\ 0.2 :: b. \\ 0.3 :: c. \\ d : -b. \\ d : -b, c. \end{array}$$

Based on the differences between $\mathcal{M}_a$ and $\mathcal{M}_h$, the generic search algorithm considers the following first-level action types (Line 16 in Algorithm 1):

$$\mathcal{A}_{\text{type}} = \{\text{change-probability, add-fact, add-rule, delete-rule}\}.$$

For each action type $a_t \in \mathcal{A}_{\text{type}}$, the corresponding candidate set $\mathcal{A}_t^{a_t}$ (Line 17 in Algorithm 1) is constructed according to the criteria described in Section 5.1. In this example, the resulting candidate sets are:

- **Change-probability:** $\mathcal{A}_t^c = \{a, b, c\}$, corresponding to facts shared by $\mathcal{M}_a$ and $\mathcal{M}_h$ but assigned different probabilities;
- **Add-fact:** $\mathcal{A}_t^a = \{e\}$, corresponding to facts present in $\mathcal{M}_a$ but absent from $\mathcal{M}_h$;
- **Add-rule:** $\mathcal{A}_t^{r,+} = \{d : -a, b.,\ d : -a, c.\}$, corresponding to rules in $\mathcal{M}_a \setminus \mathcal{M}_h$ with bodies over existing human facts;
- **Delete-rule:** $\mathcal{A}_t^{r,-} = \{d : -b.,\ d : -b, c.\}$, corresponding to rules that appear in $\mathcal{M}_h$.

This example illustrates how the action space is systematically derived from the structural and probabilistic differences between the agent and human models, while the underlying search procedure defined in Algorithm 1 remains generic and independent of the specific models considered.

## A.2 Proof of Theorem 1

*Proof.* The search algorithm exhaustively explores all possible combinations of explanation components. In particular, it will consider the explanation $\epsilon = \langle \epsilon^+, \epsilon^- \rangle$ where $\epsilon^+ = \mathcal{M}_a \setminus \mathcal{M}_h$ and $\epsilon^- = \mathcal{M}_h \setminus \mathcal{M}_a$. Applying this explanation yields the updated human model:

$$\mathcal{M}_h^* = (\mathcal{M}_h \cup \epsilon^+) \setminus \epsilon^- = \mathcal{M}_a.$$

Since $\mathcal{M}_h^* = \mathcal{M}_a$, the models are fully aligned, and thus consistent with respect to $q$. Therefore, the existence of at least one valid explanation is guaranteed. $\square$

## A.3 Proof of Theorem 2

*Proof.* We provide the proof for Case 1. The proof for Case 2 proceeds in a similar manner.

$(\Rightarrow)$ $P(\text{MPE}(q \mid \mathcal{M})) \geq P(\text{MPE}(\neg q \mid \mathcal{M})) \Rightarrow \exists i \in [m],\ \forall j \in [k_i],\ P(a_i^j) \geq 0.5$

We prove by contradiction. Suppose $P(\text{MPE}(q \mid \mathcal{M})) \geq P(\text{MPE}(\neg q \mid \mathcal{M}))$ but $\forall i \in [m], \exists j \in [k_i],\ P(a_i^j) < 0.5$.

Let $\mathcal{C}(q) = \text{MPE}(q \mid \mathcal{M})$ be the most probable explanation under which $q$ holds. Since $q = \bigvee_{i=1}^m r_i$, $q$ holds under $\mathcal{C}(q)$ if and only if at least one $r_i$ is true. Let $r_1$ be one such clause satisfied in $\mathcal{C}(q)$.

By assumption, in $r_1$ there exists some $j \in [k_1]$ such that $P(a_1^j) < 0.5$. Without loss of generality, assume $j = 1$, i.e., $P(a_1^1) < 0.5$. Construct a new explanation $\mathcal{C}'$ by setting $a_1^1$ to false, and keeping all other assignments unchanged.

Since $P(a_1^1) < 0.5$, it follows that:

$$P(\mathcal{C}') > P(\mathcal{C}(q)).$$

We now show that $q$ does not hold under $\mathcal{C}'$. Since $a_1^1$ is false in $\mathcal{C}'$, the clause $r_1$ is no longer satisfied. If $q$ were still true under $\mathcal{C}'$, then some other $r_i$ must be satisfied. But this would mean that $\mathcal{C}'$ is a valid explanation for $q$ with higher probability than $\mathcal{C}(q)$, contradicting the definition of $\mathcal{C}(q)$ as the most probable explanation of $q$.

Therefore, $q$ does not hold under $\mathcal{C}'$, i.e., $\mathcal{C}'$ satisfies $\neg q$. This leads to a contradiction:

$$P(\text{MPE}(q \mid \mathcal{M})) = P(\mathcal{C}(q)) < P(\mathcal{C}') \leq P(\text{MPE}(\neg q \mid \mathcal{M})).$$

Hence, the assumption must be false.

$(\Leftarrow)$ $\exists i \in [m],\ \forall j \in [k_i],\ P(a_i^j) \geq 0.5 \Rightarrow P(\text{MPE}(q \mid \mathcal{M})) \geq P(\text{MPE}(\neg q \mid \mathcal{M}))$

Suppose $\exists i \in [m], \forall j \in [k_i],\ P(a_i^j) \geq 0.5$. Without loss of generality, assume this holds for $r_1$.

Let $\mathcal{C}(\neg q) = \mathrm{MPE}(\neg q \mid \mathcal{M})$ be the most probable explanation under which $q$ does not hold. Since $q = \bigvee_{i=1}^{m} r_i$, $\neg q$ holds under $\mathcal{C}(\neg q)$ only if all $r_i$ are false. In particular, $r_1$ must be false under $\mathcal{C}(\neg q)$, meaning that at least one literal in $r_1$ is assigned false. Let:

$$A = \{a_1^j \mid a_1^j \text{ is false in } \mathcal{C}(\neg q), \ j \in [k_1]\}.$$

Construct a new explanation $\mathcal{C}'$ by flipping the truth values of all $a_1^j \in A$ to true, and keeping all other assignments unchanged.

Since each $P(a_1^j) \geq 0.5$, this modification leads to:

$$P(\mathcal{C}') \geq P(\mathcal{C}(\neg q)).$$

Furthermore, since $r_1 = \bigwedge_{j=1}^{k_1} a_1^j$ and all $a_1^j$ are now true in $\mathcal{C}'$, we have that $r_1$ is satisfied in $\mathcal{C}'$, and hence $q$ holds.

This yields:

$$P(\mathrm{MPE}(\neg q \mid \mathcal{M})) = P(\mathcal{C}(\neg q)) \leq P(\mathcal{C}') \leq P(\mathrm{MPE}(q \mid \mathcal{M})).$$

$\square$

### A.4 Details of Optimized Search Algorithm

The pseudocode of the optimized search algorithm follows the same overall structure as the generic search algorithm shown in Algorithm 1, differing only in the definition of the action space and the heuristic function, which are specified in Section 5.2. In addition, the INCONSISTENT predicate is evaluated according to Theorem 2. We illustrate the resulting action space and heuristic function through two examples corresponding to Case 1 and Case 2 in Definition 1.

**Case 1** In this case, the agent model supports the query $d$ while the human model does not, i.e., $P(\mathrm{MPE}(d \mid \mathcal{M}_a)) > P(\mathrm{MPE}(\neg d \mid \mathcal{M}_a))$ but $P(\mathrm{MPE}(d \mid \mathcal{M}_h)) < P(\mathrm{MPE}(\neg d \mid \mathcal{M}_h))$. The goal is to strengthen the human model so as to increase the likelihood of $d$.

$$
\mathcal{M}_a : \quad
\begin{array}{l}
0.1 :: a. \\
0.6 :: b. \\
0.7 :: c. \\
0.8 :: e. \\
d : -c, e. \\
d : -a, b. \\
d : -a, c.
\end{array}
\qquad
\mathcal{M}_h : \quad
\begin{array}{l}
0.2 :: a. \\
0.2 :: b. \\
0.3 :: c. \\
d : -b. \\
d : -b, c.
\end{array}
$$

We first specify the available first-level action types:

$$\mathcal{A}_{\text{type}} = \{\text{change-probability, add-fact, add-rule}\},$$

where the *delete-rule* action is pruned under the optimized setting for Case 1.

Based on these action types, the resulting pruned candidate action sets are:

- **Change-probability:** $\mathcal{A}_t^c = \{b, c\}$, since $P_a(b) = 0.6 > 0.5$ while $P_h(b) = 0.2 < 0.5$, and $P_a(c) = 0.7 > 0.5$ while $P_h(c) = 0.3 < 0.5$.
- **Add-fact:** $\mathcal{A}_t^a = \{e\}$, which coincides with the candidate set produced by the generic search algorithm.
- **Add-rule:** $\mathcal{A}_t^{r,+} = \emptyset$, since the current belief state $\mathcal{B}_t = \{\neg a, \neg b, \neg c\}$ (Equation 8) does not support the body of any rule.

We next illustrate the computation of the heuristic function. For each rule $r$ in the human model, we consider the set of modifiable literals $\mathcal{C}_t^r$ defined in Equation 12. In particular, for $r_1 : d :- b.$, $\mathcal{C}_t^{r_1} = \{b\}$, and for $r_2 : d :- b, c.$, $\mathcal{C}_t^{r_2} = \{b, c\}$. Thus, the minimum number of required modifications across all rules is 1.

Since $\mathcal{A}_t^{r,+} = \emptyset$, no *add-rule* action is applicable at timestep $t$. Accordingly, the optimized heuristic value $h_t^{\text{opt}}$ (Equation 13) is

$$h_t^{\text{opt}} = \min\left(c_p, \ c_r^+ + c_f^+, \ c_r^+ + c_p\right).$$

**Case 2** This example illustrates the opposite scenario, in which the human model supports the query $d$, while the agent model does not, i.e., $P(\text{MPE}(d \mid \mathcal{M}_a)) < P(\text{MPE}(\neg d \mid \mathcal{M}_a))$, but $P(\text{MPE}(d \mid \mathcal{M}_h)) > P(\text{MPE}(\neg d \mid \mathcal{M}_h))$. The goal is to weaken the human model so that $d$ becomes less probable.

$$
\mathcal{M}_a : \begin{array}{l} 0.2 :: a. \\ 0.2 :: b. \\ 0.3 :: c. \\ d : -b. \\ d : -b, c. \end{array}
\qquad
\mathcal{M}_h : \begin{array}{l} 0.1 :: a. \\ 0.6 :: b. \\ 0.7 :: c. \\ 0.8 :: e. \\ d : -c, e. \\ d : -a, b. \\ d : -a, c. \end{array}
$$

We first specify the available action types at the first level:

$$
\mathcal{A}_{\text{type}} = \{\text{change-probability, delete-rule}\},
$$

where the *add-rule* and *add-fact* actions are pruned under the optimized setting in Case 2.

Based on these action types, the pruned candidate action sets are given as follows:

- **Change-probability:** $\mathcal{A}_t^c = \{b, c\}$, since $P_a(b) = 0.2 < 0.5$ while $P_h(b) = 0.6 > 0.5$, and $P_a(c) = 0.3 < 0.5$ while $P_h(c) = 0.7 > 0.5$.
- **Delete-rule:** $\mathcal{A}_t^{r,-} = \{d :- c, e.\}$, since the opposite belief $\mathcal{B}_t' = \{a, \neg b, \neg c, \neg e\}$ (Equation 10) does not support the body of this rule.

Next, we illustrate the computation of the heuristic function under this setting. The universe of unsatisfied rules is $\mathcal{U}_t = \{d :- c, e.\}$ (Equation 14). For each modifiable fact in $\mathcal{A}_t^c$, the corresponding covering subsets are $\mathcal{S}_t^c = \{d :- c, e.\}$ and $\mathcal{S}_t^b = \{d :- a, b.\}$ (Equation 15). Consequently, the smallest collection of subsets that covers $\mathcal{U}_t$ is $\mathcal{S}_t^* = \{\mathcal{S}_t^c\}$ (Equation 16).

Since $\mathcal{S}_t^* \neq \emptyset$, the optimized heuristic value $h_t^{\text{opt}}$ defined in Equation 17 is given by

$$
h_t^{\text{opt}} = \min\big(\min(c_r^-, c_p), \ c_r^-\big).
$$

### A.5 Proof of Theorem 3

*Proof.* We provide the proof below.

**Admissibility of $h_t^{\text{gen}}$.** The generic heuristic $h_t^{\text{gen}}$ defined in Equation 6 is admissible. At timestep $t$, the current human model is still inconsistent with the agent model; hence, at least one further action is required to reach a consistent model. Since $h_t^{\text{gen}}$ only counts such necessary remaining actions, it never overestimates the optimal explanation cost and therefore is admissible.

**Admissibility of $h_t^{\text{opt}}$.** The optimized heuristic $h_t^{\text{opt}}$, defined in Equations 13 and 17, is also admissible. At timestep $t$, $h_t^{\text{opt}}$ is derived by considering all valid action types and lower-bounding the minimum cost required to eliminate the set of unsatisfied rules $\mathcal{U}_t$ (Equation 14). As shown in Section 5.2, this corresponds to a relaxed set-cover formulation, whose optimal value provides a lower bound on the true explanation cost.

In Case 2, we further restrict the rule coverage to rules with the smallest body size,

$$
\mathcal{U}_t' = \{\, r \in \mathcal{U}_t \mid |\text{body}(r)| = \min_{r' \in \mathcal{U}_t} |\text{body}(r')| \,\}.
$$

Recall that $\mathcal{S}_t^*$ denotes the minimum collection of subsets covering $\mathcal{U}_t$ (Equation 15), and let $\mathcal{S}_t^{*'}$ denote the minimum collection of subsets covering $\mathcal{U}_t'$.

We consider the following cases:

- If $\mathcal{S}_t^* = \emptyset$ and $\mathcal{S}_t^{*'} = \emptyset$, then the heuristic value remains unchanged.
- If $\mathcal{S}_t^* \neq \emptyset$ and $\mathcal{S}_t^{*'} \neq \emptyset$, then $\mathcal{U}_t' \subseteq \mathcal{U}_t$ implies $|\mathcal{S}_t^{*'}| \leq |\mathcal{S}_t^*|$, and hence the restricted heuristic does not exceed the original heuristic value.

- If $\mathcal{S}_t^* = \emptyset$ and $\mathcal{S}_t^{*'} \neq \emptyset$, then deleting at least one rule is necessary. After such deletions, the resulting universe admits a non-empty cover, and again we have $|\mathcal{S}_t^{*'}| \leq |\mathcal{S}_t^*|$.

The remaining case $\mathcal{S}_t^* \neq \emptyset$ and $\mathcal{S}_t^{*'} = \emptyset$ is impossible since $\mathcal{U}_t' \subseteq \mathcal{U}_t$. Therefore, restricting the universe to $\mathcal{U}_t'$ cannot increase the heuristic value, and $h_t^{opt}$ remains admissible.

**Guarantee for $h_t^{\mathrm{greedy}}$.** In Case 2, we formulate a relaxed set-cover instance where the universe is the set of currently unsatisfied rules $\mathcal{U}_t$ (Equation 14), and each modifiable literal induces a subset of $\mathcal{U}_t$ that it can "cover" (Equation 15). Recall that $\mathcal{S}_t^*$ denotes an optimal set cover for $\mathcal{U}_t$ (Equation 16), and let $\mathcal{S}_t^{greedy}$ denote the collection returned by the standard greedy set-cover algorithm.

In this way, the greedy set-cover solution satisfies

$$|\mathcal{S}_t^{greedy}| \leq (1 + \ln |\mathcal{U}_t|) \cdot |\mathcal{S}_t^*|.$$

Hence, the greedy heuristic achieves a $(1 + \ln |\mathcal{U}_t|)$-approximation, i.e.,

$$h_t^{greedy} \leq (1 + \ln |\mathcal{U}_t|) \cdot h_t^{opt}.$$

$\square$

## A.6 Additional Analysis of Results in Table 2

**Impact of Complexity on Runtime.** Higher complexity implies more discrepancies in probabilistic facts but fewer rules in the human model. In **Case 1**, runtime increases with complexity, as the action space for *change-probability* and *add-fact* expands, while *delete-rule* actions are not applicable. In **Case 2**, for *Generic Search*, higher complexity leads to shorter runtimes because fewer rules reduce the number of costly *delete-rule* actions, allowing A* to focus on lower-cost alternatives. In contrast, under *Optimized Search*, runtime increases with complexity, since a larger number of differing facts enlarges the search space, making the heuristic-guided A* search more expensive (see Equation 17).

**Impact on Explanation Cost.** As the model size increases, the explanation cost decreases in **Case 1** but increases in **Case 2**. In **Case 1**, a larger human model with more facts and rules enables the use of lower-cost *change-probability* actions. Moreover, by Theorem 2, it suffices for a single rule to satisfy the MPE condition, making low-cost explanations more likely. In **Case 2**, a larger human model introduces more rules that must be disabled, resulting in higher explanation costs due to the reliance on *delete-rule* actions.

At a fixed model size, increasing complexity raises the explanation cost in **Case 1** when the model is small, as more fact discrepancies must be aligned. However, when the model size is large, the cost remains stable, since most cases can be resolved by adjusting a single probability or adding a single rule, again by Theorem 2. In **Case 2**, higher complexity reduces the explanation cost, as the number of rules in the human model decreases, thereby requiring fewer costly *delete-rule* actions.

