# OpenReview forum: "Model Reconciliation via Cost-Optimal Explanations in Probabilistic Logic Programming"
_NeurIPS.cc/2025/Conference — NeurIPS 2025 poster_

### Official Review · Reviewer_y1CM · 2025-06-14

**Clarity:** 3
**Significance:** 2
**Originality:** 3
**Rating:** 4
**Confidence:** 3

**Summary:**

This paper introduces a probabilistic model reconciliation framework for explainable AI using PLP. The work addresses a significant gap in existing model reconciliation approaches, which predominantly assume deterministic models and fail to account for the probabilistic nature of human beliefs. The authors propose a framework built on ProbLog that resolves inconsistencies in MPE outcome probabilities between an agent's and a human's models. They formalize explanations as cost-optimal model updates and develop two search algorithms—a generic exhaustive search and an optimized version guided by theoretical insights. The approach is validated through a user study examining cognitive costs of different explanation types and computational experiments demonstrating the optimized algorithm's superior performance.

**Questions:**

- How does the approach scale to real-world domains with hundreds or thousands of probabilistic facts? Can you provide analysis of the theoretical complexity bounds and discuss potential approximation strategies for large-scale problems?

- The framework assumes human models are consistent and well-calibrated. How would the approach handle inconsistent human beliefs or miscalibrated probabilities? What modifications would be needed to accommodate these realistic scenarios?

- The current framework focuses on MPE-based inconsistencies. How would the approach extend to other forms of probabilistic disagreement? Would the theoretical results still hold?

- While the user study provides initial cost estimates, how sensitive are the results to different cost functions? Have you considered learning personalized cost functions or adapting costs based on user feedback during interaction?

- How does your approach compare with adapting existing deterministic model reconciliation methods to handle probability distributions? What are the fundamental advantages of the PLP-based approach over such adaptations?

**Ethical Concerns:**

["NO or VERY MINOR ethics concerns only"]

**Final Justification:**

The authors addressed most of my key concerns with clear technical clarifications. However, the introduction of new heuristic variants during the rebuttal phase without providing full methodological details affects the reproducibility and transparency of the work. Additionally, the limited number of participants and the use of a single domain in the experiments raise concerns about the generalizability of the results. Scalability issues persist as evidenced by timeouts in larger configurations, which impact the practical applicability of the method. Despite these unresolved issues, the authors’ responsiveness and the technical contributions presented warrant a positive score, though I encourage further improvements in methodological detail and experimental scope in the final version.

**Limitations:**

The authors adequately address several limitations in the conclusion, including the logic-based nature limiting real-world testing, assumptions about human model consistency, and scalability challenges. However, the discussion could be strengthened by:

1. Providing more specific bounds on when the approach becomes computationally intractable

2. Discussing concrete approaches to handle inconsistent human models

3. Addressing how the framework might extend beyond ProbLog to other probabilistic reasoning systems

4. While mentioned, the discussion of potential manipulation through adaptive explanations could be more detailed


The authors appropriately acknowledge that flawed agent models may reinforce incorrect beliefs and discuss the need for transparency and ethical design.

**Paper Formatting Concerns:**

The paper generally follows NeurIPS formatting guidelines well. Minor issues include:

1. Some mathematical notation could be more consistent (e.g., use of subscripts and superscripts)

2. Table 2 is dense and could benefit from better spacing or splitting across pages

3. The appendix proofs could be formatted more clearly with better theorem numbering

**Quality:**

2

**Strengths And Weaknesses:**

**Strengths:**

1. This represents the first work to address model reconciliation in probabilistic settings using PLP. The integration of uncertainty into model reconciliation is novel and addresses a genuine limitation in existing deterministic approaches. The cost-based explanation framework with different modification types (change-probability, add-fact, add-rule, delete-rule) is new.

2. The paper demonstrates strong technical rigor with well-formulated definitions, theorems with complete proofs, and a principled approach to cost optimization. The theoretical foundation is solid.

3. The paper is generally well-written with clear mathematical formulations. The progression from background through problem formulation to algorithms is logical. Examples effectively illustrate key concepts, particularly Example 1 and Example 2.

4. The work tackles an important problem in human-AI interaction where probabilistic beliefs are prevalent. The framework has potential applications across domains requiring explainable AI with uncertain reasoning.


**Weaknesses:**

1. The evaluation relies heavily on a controlled user study with a simple warehouse scenario. Real-world validation is absent, limiting confidence in practical applicability. The user study, while methodologically sound, involves only 100 participants and a single domain.

2. Despite the optimized algorithm's improvements, timeout issues persist in larger configurations. The exponential nature of the search space remains a fundamental limitation.

3. The framework assumes human models are consistent and well-calibrated, which may not hold in practice. The restriction to MPE-based inconsistencies may not capture all relevant forms of probabilistic disagreement.

4. The paper lacks comparison with alternative approaches to probabilistic explanation generation or adaptation of existing deterministic methods to probabilistic settings.

---

> ### Author Rebuttal · Authors · 2025-07-30
>
> We thank the reviewers for their thoughtful feedback and careful evaluation. Below, we provide detailed responses to the key concerns raised.
>
> _**Q1:**_ Please see our response to  _**Q1**_ of _**Reviewer azj8**_ for scalability to hundreds or thousands of facts and approximation approaches.
>
> Regarding _complexity_, we clarify that computing a cost-optimal explanation that aligns the agent and human models under MPE semantics is NP-hard, due to the combinatorial nature of model changes and their interactions. Even verifying the optimality of a given explanation requires comparing it against an exponential number of possible alternatives. Moreover, computing an MPE under probabilistic logic programming is known to lie in the $\Sigma_2^P$ complexity class, which is harder than NP-complete and requires reasoning via NP oracles. We will clarify this complexity in the revised version.
>
> _**Q2:**_ Our framework assumes that the **deterministic facts** in the human model are _logically consistent_, meaning atoms defined without a probability (as described in Section 3.1). These facts represent hard constraints that must always hold. For example, stating both $\texttt{bird(tweety).}$ (Tweety is a bird) and $\texttt{\\\\+bird(tweety).}$ (Tweety is not a bird) creates a direct contradiction. Including both in the same model makes it logically inconsistent, which may cause ProbLog to fail during grounding or inference. Therefore, deterministic knowledge is required to be internally consistent.
>
> In contrast, our framework can naturally handle _**inconsistencies in probabilistic beliefs**_, which are common in reasoning. For instance, a model might assign high probabilities to both $\texttt{0.9::rain.}$ (it's very likely to rain) and $\texttt{0.8::\\\\+rain.}$ (it's also very likely not to rain). While such beliefs appear contradictory, ProbLog treats them as _independent_ probabilistic events and can still compute meaningful outcomes (e.g., MPE).
>
> In summary, while the deterministic part of the model must be consistent, our framework is designed to tolerate uncertainty and soft conflicts in probabilistic beliefs, thanks to the flexible semantics of probabilistic logic.
>
> _**Q3:**_ Thank you for the insightful question. While our current framework is specifically designed to identify and resolve MPE-based inconsistencies, the Generic approach is flexible and can be adapted to handle other forms of probabilistic disagreement, e.g., MAP-based queries or general threshold-based conditions (e.g., why $P(\varphi) > \theta$).
>
> However, it is important to note that the Optimized method presented in the paper is tailored specifically for MPE computations, due to the unique structure and properties of MPE-based reasoning. These optimizations cannot be directly applied to other forms of probabilistic disagreement and would require substantial redesign for the new query type.
>
> We chose to focus on MPE-based inconsistencies because MPE captures what the model believes is the most likely complete picture of the world, rather than marginal probabilities over individual events. This type of explanation is more concrete and easier to interpret, especially when comparing how two models differ in their reasoning. It also aligns with how people naturally think since we often make decisions based on what we believe is most likely to happen. From the perspective of alignment, interpretability, and human-centered reasoning, MPE offers a clear and practical way to identify meaningful disagreements between the agent and the human.
>
> _**Q4:**_ Yes, explanation quality is sensitive to the underlying cost function, as it directly shapes which explanations are considered optimal. In our current work, we use cost estimates derived from a human-subject study in a smart warehouse domain to reflect how users perceive the cognitive effort of different explanation types. We do not assume these costs generalize across all domains or users, but our framework is flexible since new cost models can be substituted when additional preference data becomes available.
>
> If a user’s preference costs are known in advance (e.g., from prior interactions or user profiles), they can be directly incorporated to generate more personalized, cognitively aligned explanations.
>
> As a next step, we believe it is promising to adapt cost models dynamically through user interaction, as done in deterministic model reconciliation work that progressed from one-shot [1] to multi-shot [2] and learning-based [3] formulations. Our framework is compatible with such extensions and can support personalized reconciliation under uncertainty in future interactive settings.
>
> _**Q5:**_ A direct adaptation of deterministic reconciliation methods is not sufficient to handle uncertainty in probabilistic settings. In deterministic cases, differences between the agent and the human are typically binary. For example, they may disagree on whether an action is allowed or whether a condition holds. In contrast, probabilistic settings involve more subtle forms of disagreement. The agent and the human may both agree that something can happen, but they may differ in how likely they think it is or how strongly they believe a particular fact or rule. These are not simple yes-or-no differences; they reflect varying degrees of belief. As a result, we need more fine-grained, numerical explanations that deterministic methods are not equipped to provide. To address this, we develop a probabilistic model reconciliation framework that can capture and resolve such belief discrepancies.
>
> PLP offers a powerful way to represent uncertain knowledge in a structured and interpretable manner. It combines the expressiveness of logic with probabilistic reasoning, making it well suited for modeling belief differences between agents and humans. Unlike black-box statistical models, PLP provides a transparent framework for describing beliefs and how different facts and rules interact, making the reasoning process easier to understand and explain. Moreover, PLP supports inference tasks such as Most Probable Explanation (MPE) and Maximum a Posteriori (MAP), which align directly with our goal of generating concise and meaningful explanations under uncertainty. We will clarify this motivation further in the revised version.
>
> _**References**_
>
> [1] "Model Reconciliation in Logic Programs." Son et al., in Proceedings of JELIA, 2021.
>
> [2] "Dialectical Reconciliation via Structured Argumentative Dialogues." Vasileiou et al., in Proceedings of KR, 2024.
>
> [3] "Does Your AI Agent Get You? A Personalizable Framework for Approximating Human Models from Argumentation-based Dialogue Traces." Tang et al., in Proceedings of AAAI, 2025.

---

> > ### Comment · Reviewer_y1CM · 2025-08-05
> >
> > Thank you for your detailed response and for addressing several of my concerns.
> >
> > Regarding my earlier question, my main concern was about the experimental setup involving only 100 participants within a single domain. This raises questions about the generalizability of the findings, which I feel remains insufficiently addressed.
> >
> > I also note that timeouts still occur for larger configurations, which impacts the method’s scalability in practice. Except for the newly presented methods (which uses the enhanced heuristic functions developed after submission).
> >
> > Overall, you have addressed most of my other queries thoughtfully, and I will keep a positive score in my evaluation. I encourage you to provide these additional details and clarifications in the final version to further strengthen the work.

---

> > > ### Author Response · Authors · 2025-08-06
> > >
> > > Thank you for acknowledging our responses and for the positive evaluation. We appreciate your thoughtful engagement and will incorporate your suggestions to further strengthen the final version.
> > >
> > > Below, we provide further clarifications on the remaining concerns, specifically regarding the generalizability of the findings and the scalability of our method in larger instances.
> > >
> > > _**Q1: Generalizability of the Findings**_
> > >
> > > A1: While our user study involved 100 participants within a single domain, we would like to emphasize that this setup does **NOT** limit the general applicability of our framework.
> > >
> > > The primary purpose of the study was to demonstrate a _feasible and practical_ method for eliciting human-aligned cost values, which are inherently subjective and vary across users. These costs were used to instantiate our cost-optimal explanation generation procedure. _**Importantly, the specific numerical values of the costs do NOT affect the theoretical foundations or computational properties of our method.**_ They only influence which explanation is ultimately selected as optimal, without impacting the underlying algorithmic process or its guarantees.
> > >
> > > Therefore, the findings of the study are not constrained by the particular domain or the size of the participant pool. Our framework is inherently domain-independent, and in future work, it can incorporate user-specific cost profiles derived from prior interactions.
> > >
> > > _**Q2: Timeout Problems**_
> > >
> > > A2: Since computing a cost-optimal explanation that aligns the agent and human models under MPE semantics is **NP-hard**, we propose using the newly introduced heuristic function $h_{greedy}$ to obtain suboptimal solutions that remain within provable theoretical bounds, especially for larger problem instances.

---

### Official Review · Reviewer_mmqa · 2025-07-02

**Clarity:** 3
**Significance:** 2
**Originality:** 2
**Rating:** 4
**Confidence:** 4

**Summary:**

In this paper, the authors consider the problem of reconciling two
models that are specified as probabilistic logic programs. More
precisely, the authors consider the notion of most probable
explanation (MPE), which, for a query q and a probabilistic logic
program M, is denoted by MPE(q | M). Then, two probabilistic logic
programs M_1 and M_2 are said to be consistent with respect to a query
q if either Pr(MPE(q | M_1)) > Pr(MPE(not q | M_1)) and Pr(MPE(q |
M_2)) < Pr(MPE(not q | M_2)), or the other way around. That is, M_1
and M_2 have opposite evidence for the queries q and not q. The goal
of this paper is to solve this type of inconsistencies.

The authors provide a formal definition of what it means to solve an
inconsistency for two probabilistic logic programs M_1 and M_2,
which is based on the idea of removing and adding facts and rules in one of these
programs. They introduce a cost measure for such updates, which are
called P-MRP explanations. Moreover, they define the problem of
computing an optimal P-MRP explanation for an inconsistency between
probabilistic logic programs, that is, a P-MRP explanation of minimal
cost. Finally, they provide a generic search algorithm and an
optimized search algorithm, and an experimental evaluation showing
some advantages of the optimized version.

**Questions:**

Q1 What is the complexity of the optimized search algorithm?

**Ethical Concerns:**

["NO or VERY MINOR ethics concerns only"]

**Final Justification:**

My impression of the paper has improved after all the discussions. I am increasing my final score, although there are still some concerns about its contributions.

**Limitations:**

Yes

**Paper Formatting Concerns:**

No formatting issues

**Quality:**

2

**Strengths And Weaknesses:**

Strengths:

S1 Model reconciliation is a fundamental problem for which this paper
proposes a more principled approach based on formalizing the problem
and proposing a solution based on probabilistic logic programming.

Weaknesses:

W1 The paper works under the assumption that probabilistic logic
programming is an appropriate model to formalize user models, but it
does not provide clear evidence for this assumption.

W2 The two algorithms proposed in the paper are based on usual
techniques, and they do not provide a deep insight into how model
reconciliation should be done.

W3 The paper does not study the complexity of the problem of computing
optimal P-MRP explanations. In particular, it does not provide any
lower bounds for this problem, which could help to understand how
good the optimized search algorithm proposed in the paper is.

---

> ### Author Rebuttal · Authors · 2025-07-30
>
> Thank you for your thoughtful feedback. We appreciate your careful reading and respond to the key concerns below.
>
> _**Q1: The paper assumes PLP is suitable for modeling user models, but this is not justified.**_
>
> A1: Because both the agent and the human may have uncertain or incomplete beliefs about the world, we adopt probabilistic methods to represent this uncertainty. In particular, PLP has proven effective for modeling uncertain knowledge in a structured and interpretable way. It has been successfully applied in domains such as diagnosis [1] and decision support [2].
>
> In our work, PLP serves as a formal modeling language for representing the knowledge of both the agent and the human. The goal is **NOT** to simulate human reasoning, but rather to express the assumptions, dependencies, and uncertainties in a clear and compact way. PLP offers a good balance between expressivity, interpretability, and conciseness, making it suitable for modeling complex probabilistic domains.
>
> Moreover, PLP naturally supports inference tasks like most probable explanation (MPE) and maximum a posteriori (MAP), which align with our goal of generating concise and meaningful explanations that resolve belief differences. We will clarify this motivation in the revised version.
>
> _**Q2: No complexity or lower bound analysis is provided for computing optimal P-MRP explanations.**_
>
> A2: In particular, computing a cost-optimal explanation that aligns the agent and human models under MPE semantics is NP-hard, due to the combinatorial nature of model changes and their interactions. Even verifying the optimality of a given explanation requires comparing it against an exponential number of possible alternatives. Moreover, computing an MPE under probabilistic logic programming is known to lie in the $\Sigma_2^P$ complexity class, which is harder than NP-complete and requires reasoning via NP oracles. We will clarify this complexity in the revised version.
>
> _**Q3: The proposed algorithms use standard techniques and offer limited insight into reconciliation.**_
>
> A3: We would like to clarify that while our methods build upon the general A* framework, our contributions go _**significantly**_ beyond standard applications. We introduce a novel formulation of probabilistic model reconciliation under MPE semantics, with cost-optimal explanations over PLP-based models, and develop tailored algorithms that incorporate theoretical, algorithmic, and human-centered insights.
>
> In particular:
> * We are the _**first**_ to formally define model reconciliation under _uncertainty_, specifically targeting inconsistencies in MPE outcomes between agent and human ProbLog models.
> * While the general problem is intractable in the worst case (See A2), our framework incorporates several non-trivial strategies to ensure practical efficiency (see our response to  _**Q1**_  of  _**Reviewer azj8**_ for details):
>   - Our optimized A* search incorporates principled pruning based on theoretical insights such as DNF clause filtering and Theorem 2, which significantly reduces the effective search space compared to the generic version.
>   - We propose tighter admissible heuristics for exact-optimal search and bounded-suboptimal greedy heuristics with formal $(1 + \ln n)$-approximation guarantees. These heuristics scale efficiently, even in large models where exact A* may time out.
>
>   These improvements go beyond standard pruning techniques and enable our framework to achieve both scalability and theoretical guarantees.
> *  We also ground our cost model in a human-subject study, where explanation types (e.g., change-probability, delete-rule are evaluated via pairwise preference comparisons and modeled using the Bradley–Terry framework. This enables the system to generate explanations that are both logically valid and cognitively aligned with user preferences.
>
> Taken together, our approach contributes algorithmic, theoretical, and human-centric advances to the problem of model reconciliation under uncertainty. We will clarify and highlight these contributions more explicitly in the revised version.
>
> _**References**_
>
> [1] "SimPLoID: Harnessing Probabilistic Logic Programming for Infectious Disease Epidemiology." Weitkämper et al., arXiv preprint, 2023.
>
> [2] "Intention Recognition with ProbLog." Smith et al., in Frontiers in Artificial Intelligence, 2022.

---

### Official Review · Reviewer_abKX · 2025-07-02

**Clarity:** 3
**Significance:** 2
**Originality:** 2
**Rating:** 4
**Confidence:** 4

**Summary:**

This paper studies the model reconciliation problem as explanation generation under probabilistic logic programming (PLP). It extends the existing work on deterministic/stochastic planning models to a PLP setting where model is specified as facts and rules associated with their respective probabilities. The goal is to reconcile between two models by explaining their differences such that a specific query becomes consistent across the two models while minimizing the cost of the explanation, captured as changed probability, added fact/rule, etc. Two search methods are proposed where the first is a systemic search through the entire solution space while the second allows limited pruning to expedite the search. The proposed method is evaluated via a user study that analyzes the weights of different explanation types on the user as well as computational evaluations showing the effectiveness of the proposed search methods.

**Questions:**

1. Is there any reason why the generated explanations are not directly compared with simple baselines for effectiveness (such as explaining facts and rules that differ in the two models and are most relevant to the query)? This can be done both with human studies and via computational evaluations and would provide direct evidence to support the proposed method.

2. Can you better motivate model reconciliation as explanation generation under PLP? Why do you believe PLP is better suited for model reconciliation? Why is considering such probabilistic modeling important? I am asking these since PLP may be considered even less intuitive for human understanding than planning models, such as PDDL.

3. What are the explanations being compared in the human study? Can you explain the evaluation process in more detail? How are the costs estimated? Do you expect the estimated costs to be generalizable?

**Ethical Concerns:**

["NO or VERY MINOR ethics concerns only"]

**Final Justification:**

While I agree with the authors that model reconciliation under PLP can be a valuable contribution, especially under Bayesian theory of mind (my original comment was to encourage the authors to better connect with relevant studies in cognitive science), I do not agree with the claim that a direct evaluation with user studies is orthogonal. While reviewing the paper, I was surprised that the authors leveraged a user study to estimate weights for the different types of explanations instead of evaluating the effectiveness of the explanations (even better if evaluating with the estimated weights).

Having said that, I consider the work a good addition to the model reconciliation line of work. Trusting the authors' commitment to incorporating the additional heuristic that would alleviate the computational concern (which was however not included in the original submission), I would consider it to good contribution to NeurIPS. Hence, I have adjusted my rating to "weak accept".

**Limitations:**

The key limitation here is the lack of direct evaluation of the generated explanations, which is critical to establish the effectiveness of the proposed method. The fact that the authors are aware unfortunately does not alleviate the concern.

**Quality:**

2

**Strengths And Weaknesses:**

Strengths:

1. The paper addresses a gap in model reconciliation as explanation generation.
2. The paper is well structured and clearly written.
3. The technical contribution seems sound and useful.

Weaknesses:

1. The motivation for studying model reconciliation under PLP could be strengthened. The current motivation orients around what is not done instead of why it should be done.
2. The proposed search methods are generally intractable even with pruning.
3. The evaluation via human-user study is missing details and does not provide a convincing argument for the effectiveness of the  explanations generated in the real-world.
4. The proposed search methods are for one-shot explanation generation. Interaction of multiple explanations is not addressed, which would make it more relevant to real-world scenarios.
5. The proposed are mainly search methods that are perhaps better suited for AAAI, IJCAI, etc.

---

> ### Author Rebuttal · Authors · 2025-07-30
>
> Thank you very much for the comments! Our responses to your questions are below.
>
> _**Q1:**_ In fact, our Generic method can serve as a reasonable baseline for comparison. When explanation cost is not considered, it can produce any explanation that resolves the inconsistency, including a _**simple baseline**_ that lists differences in facts and rules between the two models that seem relevant to the query.
>
> However, as shown in Theorem 2, not all such differences are actually helpful for reconciliation. For example, changing the probability of a fact from 0.4 to 0.3 might have no effect on the MPE result, yet a simple baseline would still include it. This often leads to explanations that are overly long and include changes that make little difference, especially when there are many facts and rules in the model.
>
> Our method addresses this by focusing on explanations that are truly necessary to resolve the disagreement while keeping the cost as low as possible. This leads to explanations that are shorter, clearer, and more effective at helping the human and the agent reach the same understanding.
>
> _**Q2:**_  Because both the agent and the human may have uncertain or incomplete beliefs about the world, we adopt probabilistic methods to represent this uncertainty. In particular, PLP has proven effective for modeling uncertain knowledge in a structured and interpretable way. It has been successfully applied in domains such as diagnosis [1] and decision support [2].
>
> We use PLP to represent the knowledge held by both the agent and the human. The goal is **NOT** to simulate human reasoning but to provide a formal modeling language for representing their beliefs. Similar to how PDDL is used in planning, PLP allows us to express assumptions and dependencies in a declarative form. It offers a good balance of expressivity, interpretability, and compactness, and is well suited for modeling complex probabilistic domains, including planning under uncertainty.
>
> Moreover, PLP naturally supports inference tasks like most probable explanation (MPE) and maximum a posteriori (MAP), which align with our goal of generating concise and meaningful explanations that resolve belief differences. We will clarify this motivation in the revised version.
>
> _**Q3:**_ Please see our response to **Q2** of _**Reviewer azj8**_ for details on the types of explanations compared and the evaluation process.
>
> Regarding cost estimation, we fit a Bradley–Terry model based on pairwise human preferences. Each explanation is represented as a set of atomic model updates (e.g., change-probability, delete-rule), and participants are asked to compare pairs of explanations and select the one they find more understandable.
>
> **Illustrative Example.** Suppose 100 participants compare the following two explanations:
> * **E1**: _“Item B is highly likely to be part of a high-priority order, whereas it was initially considered much less likely.”_ (change-probability)
> * **E2**: _“The assumption that the task should only be executed when Item B is part of a high-priority order is inaccurate.”_ (delete-rule)
>
> Assume 70 participants preferred E1, and 30 preferred E2. Let the latent utility parameters for these explanation types be $\beta_{cp}$ (change-probability) and $\beta_{dr}$ (delete-rule), respectively.
> Under the Bradley–Terry model, the probability that the participant prefers one explanation over another is defined as:
> $$P(cp \succ dr)=\frac{e^{\beta_{cp}}}{e^{\beta_{cp}} + e^{\beta_{dr}}}=\frac{70}{100}.$$
>
> To fit the model, we _maximize the log-likelihood_ of observed preferences, i.e.,
> $$\log L=70 \cdot \log(\frac{e^{\beta_{cp}}}{e^{\beta_{cp}} + e^{\beta_{dr}}})+30\cdot\log( \frac{e^{\beta_{dr}}}{e^{\beta_{cp}} + e^{\beta_{dr}}})$$
>
> Once the $\beta$ values are learned, the cognitive cost of each update type $f$ is computed as: $\text{Cost}(f) = e^{-\beta_f}$.
> This formulation ensures an intuitive interpretation: explanation types with higher perceived utility (i.e., preferred more often, higher $\beta$) are assigned lower cognitive costs. These learned costs are then integrated into our reconciliation framework to guide the selection of preferred explanation types.
>
> **Generalizability.** Our current cost estimates are based on user preferences collected in a smart warehouse scenario. They reflect how people in that context perceive the effort of understanding different explanation types.
>
> We do not assume these costs apply to all domains or users. However, our framework is flexible: the cost model can be adjusted when new preference data is available. In this sense, the current estimates are just one example of how the method can be applied.
>
> In future work, we plan to study how explanation preferences vary across domains and individuals, and how to adapt explanation generation accordingly.
>
> We will clarify all these parts in the revised version.
>
> _**Q4: The proposed search methods are generally intractable even with pruning.**_
>
> A4:  We acknowledge that the overall problem is theoretically intractable. In particular, computing a cost-optimal explanation that aligns the agent and human models under MPE semantics is NP-hard, due to the combinatorial nature of model changes and their interactions. Even verifying the optimality of a given explanation requires comparing it against an exponential number of possible alternatives. Moreover, computing an MPE under probabilistic logic programming is known to lie in the $\Sigma_2^P$ complexity class, which is harder than NP-complete and requires reasoning via NP oracles. We will clarify this complexity in the revised version.
>
> While the general problem is indeed intractable in the worst case, our proposed framework incorporates several non-trivial strategies to ensure practical tractability. _**Reviewer  azj8**_ also had the same concern and we responded in detail to them (see our response to their **Q1**). We briefly describe them here again for convenience.
>
> First, the optimized A* search introduces aggressive pruning based on theoretical insights (e.g., DNF clause filtering, Theorem 2), which significantly reduces the effective search space compared to the generic version.
>
> Second, we introduce new admissible heuristics that are tighter than the original, leading to faster convergence in exact-optimal search. More importantly, we provide bounded-approximate variants using greedy heuristics with provable guarantees (within $(1+\ln n)$ of the optimal cost), which scale well even on large instances.
>
> As shown in our new results, the greedy strategies remain efficient and yield suboptimal explanations in cases where exact A* may time out. These improvements go beyond standard pruning and make the framework practically scalable. We will clarify the complexity landscape and our strategies in the revised version.
>
> _**Q5: The proposed search methods are for one-shot explanation generation. Interaction of multiple explanations is not addressed, which would make it more relevant to real-world scenarios.**_
>
> A5: Thank you for the suggestion.
> Our work represents an important _**first step**_ toward probabilistic model reconciliation by formalizing one-shot explanation generation under MPE semantics. This mirrors the trajectory of prior work in deterministic model reconciliation, which began with one-shot [3] formulations before progressing to multi-shot [4] and learning-based variants [5].
>
> Our framework is compatible with such extensions. In particular, the cost model we propose can be adapted or personalized over time based on user interaction data, enabling explanation generation to evolve dynamically across multiple rounds. We will clarify this potential for extension in the revised version.
>
> _**References**_
>
> [1] "SimPLoID: Harnessing Probabilistic Logic Programming for Infectious Disease Epidemiology." Weitkämper et al., arXiv preprint, 2023.
>
> [2] "Intention Recognition with ProbLog." Smith et al., in Frontiers in Artificial Intelligence, 2022.
>
> [3] "Model Reconciliation in Logic Programs." Son et al., in Proceedings of JELIA, 2021.
>
> [4] "Dialectical Reconciliation via Structured Argumentative Dialogues." Vasileiou et al., in Proceedings of KR, 2024.
>
> [5] "Does Your AI Agent Get You? A Personalizable Framework for Approximating Human Models from Argumentation-based Dialogue Traces." Tang et al., in Proceedings of AAAI, 2025.

---

> > ### Comment · Reviewer_abKX · 2025-08-04
> >
> > Thank you for the detailed response. Here is my general feeling after the rebuttal:
> >
> > Concerns remained:
> >
> > 1. The motivation to use PLP for humans remains to be strengthened. For instance, it is unclear whether normal users would even understand such explanations.
> > 2. No direct evaluation of the generated explanations and/or comparisons with simple alternatives.
> >
> > Concerns alleviated:
> >
> > 1. Thank you for the explanation regarding the cost. Very helpful. One comment is that the perceived utility may not be solely due to the cognitive cost since the quality would also be subject to assessment by the users.
> > 2. Computational complexity: You discussed about a new admissible heuristic. However, I could not find any reference in the paper. Is it something introduced AFTER the submission?

---

> > > ### Author Response · Authors · 2025-08-04
> > >
> > > Thank you for your suggestions. Our responses to your concerns are below.
> > >
> > > _**Q1: The motivation to use PLP for humans remains to be strengthened. For instance, it is unclear whether normal users would even understand such explanations.**_
> > >
> > > A1: Thank you for the comment. We would like to clarify that we use PLP primarily as _a representation to model the agent's and human’s uncertain or incomplete beliefs_ about the world. However, we _do not expect users to understand PLP in its logical form._
> > >
> > > In our user studies, all explanations are translated into natural language. For example, a PLP rule such as
> > > $$\texttt{deliver(Item) :- urgent(Item), available(Truck)}.$$
> > > would be verbalized as: "The item will be delivered if it is urgent and a truck is available.” Even when the underlying action in our generated explanation involves _deleting a rule_, the explanation shown to the user is phrased more naturally, such as: "The assumption that the item will be delivered if it is urgent and a truck is available is inaccurate".
> > >
> > > This translation bridges formal reasoning and human understanding, preserving the uncertainty encoded in PLP while making the explanation accessible to human users.
> > >
> > > In addition, we anticipate that LLMs will become increasingly useful for supporting this translation process, especially as they become more robust against hallucinations and better grounded in symbolic reasoning. For example, TRACE-cs [1] presents a novel hybrid system that combines symbolic reasoning with LLMs to generate natural language explanations for scheduling problems. We will update the revised version to better reflect this direction.
> > >
> > >
> > > _**Q2: No direct evaluation of the generated explanations and/or comparisons with simple alternatives.**_
> > >
> > > A2: Thank you for the comment. We would like to clarify that the **generic** algorithm used in our paper can serve as the _baseline_ method. When explanation cost is not considered, this algorithm can generate an explanation that lists all differences in facts and rules between the agent and human models that appear relevant to the query, referred to as **simple diff**. While such an explanation can technically resolve the inconsistency, it is often unnecessarily long and redundant, especially when the number of facts and rules is large.
> > >
> > > In our human-user study, participants consistently preferred explanations that are shorter and easier to understand. _This supports our motivation for generating cost-optimal explanations, which aim to identify the smallest and most relevant changes_. For this reason, we focus our comparison on the optimized method versus the generic baseline in terms of computational running time, and do not separately compare against a simple diff in terms of the generated explanation.
> > >
> > > Since this is the _**first**_ work on probabilistic model reconciliation, there are **NO** other existing algorithms for this problem. We will clarify this point in the revised version.
> > >
> > > _**Q3: One comment is that the perceived utility may not be solely due to the cognitive cost since the quality would also be subject to assessment by the users.**_
> > >
> > > A3: Thank you for the note. We appreciate this perspective and will include a sentence to reflect it more clearly in the revised version.
> > >
> > > _**Q4: Computational complexity: You discussed about a new admissible heuristic. However, I could not find any reference in the paper. Is it something introduced AFTER the submission?**_
> > >
> > > A4: Yes, the new admissible heuristics were introduced after the initial submission as part of our effort to further strengthen the computational analysis on scalability. We added these heuristic functions to demonstrate that our framework can help reduce search time in practice, especially on larger problem instances, while still preserving optimality guarantees.
> > >
> > > We emphasize that this addition is intended to complement the original submission and does not affect the core framework, main contributions, or theoretical claims of the paper. We will include these in our revised version.
> > >
> > > _**References**_
> > >
> > > [1] "TRACE-CS: A Synergistic Approach to Explainable Course Scheduling Using LLMs and Logic." Vasileiou et al., in Proceedings of AAAI, 2025.

---

> > > > ### Comment · Reviewer_abKX · 2025-08-07
> > > >
> > > > Thank you for the response. I will explicate more on my points above:
> > > >
> > > > After offering an explanation, the assumption is that a user would need to follow process that is a similar to the inference process of PLP to make sense of the explanation. It helps when you convert the explanation into natural languages, but ultimately the inference process (to understand) is what users need to go through. It is critical to make sure such understanding actually occurs since otherwise we can always just offer placebos such as "No worries. everything is under control."
> > > >
> > > > Hence, to show that an explanation strategy is useful, ideally, we will need to 1) motivate it by providing evidence that such explanations and inference process are psychological studied and well suited for human reasoning, and 2) provide empirical evidence for the "effectiveness" of the explanations (i.e., users would no longer make mistakes that they would have made before they corrected their belief), and that the explanations are preferred over simple strategies such as offering placebos or simple diff.
> > > >
> > > > Assuming my understanding is correct, there is no direct evaluation of the effectiveness of the explanations via human studies. The current study is used to generate preferences towards the different types of explanations used in the proposed approach.
> > > >
> > > > Regarding the new heuristic, I am not sure whether we as the reviewers should consider it since it is not in the current submission. We may need the AC's opinion on this.

---

> > > > > ### Author Response · Authors · 2025-08-08
> > > > >
> > > > > Thank you so much for the thoughtful comment and detailed justifications.
> > > > >
> > > > > In this work, we focus on generating explanations that address the problem of model reconciliation under uncertainty. Our approach is grounded in the view that Bayesian inference (e.g., MAP, MPE) captures key aspects of human cognition, particularly in how people integrate prior beliefs with observed evidence, as supported by extensive research in cognitive science [1–3]. Moreover, probabilistic inference explanations such as MPE and MAP have been studied for decades and are widely adopted in the probabilistic reasoning literature [4, 5]. We follow the established assumption in this field that such explanations are meaningful and useful for humans.
> > > > >
> > > > > Building on this foundation, we propose an explanation generation framework that targets **inconsistencies in MPE outcomes** between agent and human probabilistic models, and we develop methods to **compute model-reconciling MPEs** under formal PLP semantics.
> > > > >
> > > > > While a user study to evaluate the efficacy of such explanations would be valuable, it is **orthogonal** to our contributions, which lie in the _formal definition, theoretical analysis, and computation of model-reconciling explanations_.
> > > > >
> > > > > _**References**_
> > > > >
> > > > > [1] “Bayesian Models of Cognition.” Griffiths et al., 2008.
> > > > >
> > > > > [2] “Optimal Predictions in Everyday Cognition.”Griffiths et al., Psychological Science, 2006.
> > > > >
> > > > > [3] “Bayesian Learning Theory Applied to Human Cognition.” Jacobs et al., Wiley Interdisciplinary Reviews: Cognitive Science, 2011.
> > > > >
> > > > > [4] “Reasoning with Probabilistic and Deterministic graphical models: Exact algorithms.” Dechter Rina, 2022.
> > > > >
> > > > > [5] “Probabilistic Reasoning in Intelligent Systems: Networks of Plausible Inference.” Pearl Judea, 2014.

---

### Official Review · Reviewer_azj8 · 2025-07-06

**Clarity:** 3
**Significance:** 3
**Originality:** 3
**Rating:** 5
**Confidence:** 4

**Summary:**

This paper introduces Model Reconiliation, that is, unifying a learned model with an user model, in the context of the probabilisti logic programming language Problog, Te authors  take advantage of  MPE learning  to reduce problem complexity, implement two search algorithms and evaluate them in a dataset.

**Questions:**

Although not directly connected, the approach reminded me on the work on Theory revsion, and namely the Forte work by Mooney's group and follow-up work on the Probabilistic Forte work (Aline Paes MLJ ).

**Ethical Concerns:**

["NO or VERY MINOR ethics concerns only"]

**Limitations:**

Scalability and existing dataset come to mind.

**Paper Formatting Concerns:**

....

**Quality:**

3

**Strengths And Weaknesses:**

The authors claim to be the first to pursue model reconciliation in the context of probabilistic logic program. Reconciliation may be seem as a way to compare a learned theory with the user expectations, by  detecting and describing major differences

Problog and Model Reconciliation look like a good match, given that the probabilities may be helpful in finding the divergence between the user and learned , On the other hand, efficiency is a real problem for probabilistic logic systems, Eg, ProbLog compiles the program into trees for probability inference, but that requires the program to be stable. Using most likely explanations  seems to be a way to achieve acceptable performance. I think thatś the main contribution of the paper.

Most likely explanations avoid the complexity of computing the full probability while still generating answers  of interest.  I also think  that finding a transformation of paths instead of complex trees may be easier for a human to understand.

The problem then reduces to searching for the best transformations. The authors present a naive naive generator and ta A* algorithm, Popular and often  more effective techniquesm eg MonteCarlo search or MAXSAT should be considered.

Finally, the experimental evaluation is the weakest component of the paper. The source of the dataset should be made clear, so should the task. Experimental evaluation is hard in model reconciliation, but I would have at least expected some discussion on interpretability of the results.

My evaluation is based on the fact that I llike the idea, not just for reconciliation but  the use of MLEs. I would have expected more frrom the evaluation, though.

---

> ### Author Rebuttal · Authors · 2025-07-30
>
> Thank you for the helpful and constructive comments. We appreciate your suggestions and address them below.
>
> **_Q1: Only naive and $A^*$ are explored; other methods (e.g., Monte Carlo, MaxSAT) can be considered._**
>
> A1: Our work focuses on two A*-based methods: a generic version and an optimized one. Both are grounded in a unified framework for cost-optimal model reconciliation under MPE semantics. The optimized search is not merely a variant but introduces principled pruning based on our proposed DNF analysis (Theorem 2), resulting in substantial improvements in scalability.
>
> We agree that exploring alternative approaches could be valuable. However, incorporating MaxSAT is not straightforward in our setting. The dependencies between actions, such as changing probabilities and adding or deleting rules, are difficult to encode using propositional logic. Monte Carlo methods may help with approximate inference or serve as heuristic guidance, but aligning the sampled model updates with both MPE consistency and cost-optimality is a non-trivial challenge. We see these directions as promising future work.
>
> In the original version, both the generic and optimized methods employed the same admissible heuristic. Since submission, we have developed enhanced heuristic functions, which we now include in this revision.
> * A tighter admissible heuristic $h_{\text{exact}}$ derived from DNF pruning and set-cover estimation. This heuristic remains admissible while improving runtime compared to the original.
> * A greedy heuristic $h_{\text{greedy}}$, which solves a relaxed set-cover problem and has a $(1+\ln n)$-approximation guarantee, where $n$ is the number of uncovered DNF clauses. This ensures the total explanation cost lies within $[c^\star, (1+\ln n) c^\star]$ where $c^\star$ is the optimal cost. In practice, $n$ is often much smaller than the total number of rules, making the approach highly efficient.
>
> This enhanced heuristic improves scalability while preserving formal guarantees. Further, we exploit properties from Weighted A*, which uses weighted heuristics $w \cdot h$. Using this approach, if the heuristic used is $h_{\text{exact}}$, then the explanation cost is guaranteed to be bounded from above by $w \cdot c^\star$. When using $h_{\text{greedy}}$, the cost is guaranteed to be bounded from above by $w \cdot  (1+\ln n) c^\star$.
>
> Below, we present a comparative table of average running times for all **five** heuristic variants evaluated across different model sizes: the original heuristic (same in Generic) $h_{\text{original}}$, the new exact admissible heuristic $h_{\text{exact}}$, the greedy heuristic $h_{\text{greedy}}$, the weighted exact heuristic ($w \times h_{\text{exact}}$), and the weighted greedy heuristic ($w \times h_{\text{greedy}}$), where we set $w = 2$ in experiments.
>
> *Note: Due to space constraints, we only report results under Case 2, which involves higher complexity compared to Case 1. For each configuration, we generate 100 Agent–Human model pairs. A time limit of 600 seconds is imposed per run; “t/o” indicates a timeout, and “–” denotes that all runs timed out.*
> |Agent Facts|Agent Rules|Human Complexity|$h_{\text{original}}$(ms)|$h_{\text{exact}}$(ms)|$h_{\text{greedy}}$(ms)|$2\times h_{\text{exact}}$(ms)|$2\times h_{\text{greedy}}$(ms)|
> |:-:|:-:|:-:|:-:|:-:|:-:|:-:|:-:|
> |10|5|20%|29.1|32.2|31.0|32.7|35.6|
> |||40%|29.1|32.2|30.9|33.1|37.8|
> |||60%|29.1|32.1|31.1|34.3|35.8|
> |||80%|28.9|31.9|30.7|35.8|39.2|
> |20|10|20%|29.5|32.5|30.9|33.0|36.3|
> |||40%|29.6|32.8|30.9|32.8|36.1|
> |||60%|29.7|32.4|31.2|33.6|41.6|
> |||80%|29.9|32.6|31.5|33.9|40.0|
> |50|25|20%|77.7|32.2|33.2|33.4|36.5|
> |||40%|279.4|34.3|33.2|34.5|37.5|
> |||60%|384.7|47.7|36.1|40.5|43.6|
> |||80%|4154.6|112.0|42.4|70.9|40.8|
> |100|50|20%|37420.0 (5 t/o)|37.7|38.8|36.5|38.2|
> |||40%|98684.2 (38 t/o)|247.7|47.6|116.8|40.0|
> |||60%|124163.4 (55  t/o)|3612.2|73.6|1997.1|45.1|
> |||80%|69932.6 (47  t/o)|9658.6 (1  t/o)|139.9|3407.89 (1 t/o)|53.04|
> |1000|500|20%|-|-|104469.1 (56 t/o)|-|889.1|
> |||40%|-|-|-|-|3122.3|
> |||60%|-|-|-|-|7554.6|
> |||80%|-|-|-|-|18936.6|
>
> We observe that the new variant with $h_{\text{exact}}$ consistently achieves lower runtime compared to the original A*, while maintaining correctness. All the approximate variants with inadmissible heuristics (i.e., $h_{\text{greedy}}$, $w \times h_{\text{exact}}$, and $w \times h_{\text{greedy}}$) yield suboptimal solutions within their theoretical bounds.
> **Furthermore, in large-scale settings where the exact variant times out, the approximate variants remain tractable and effective. More importantly, this framework also allows one to use larger weights $w$ if necessary to scale to even larger problems.** We will include these results in the revised version.
>
>
> **_Q2: Experimental evaluation is weak; dataset and task need clarification. Interpretability should be discussed._**
>
> A2: ***Interpretability*** is a key focus of our work. Each explanation in our framework corresponds to a small and meaningful model update, such as changing the probability of a fact or deleting a rule. These updates have clear semantic meaning and are designed to reflect beliefs that a human can easily understand.
>
> To evaluate this, we designed a human-subject study described in Section 6.1. **The full study materials and instructions are included in the supplementary materials.**
>
> **Study Setup.** The study takes place in a smart warehouse scenario, where the AI agent, Blitzcrank, decides to execute a delivery task, which corresponds to a query in our model. However, the human operator holds conflicting prior beliefs and would have chosen not to execute the task. For example, the operator may believe:
> * _“Item A is highly likely to require additional inspection.”_ (fact-level belief)
> * _“The task should only be executed if Item B is part of a high-priority order.”_  (rule-level belief)
>
> **Explanations Provided and Compared.**
> Participants are shown candidate explanations generated by Blitzcrank to justify why the task should be executed. Each explanation corresponds to a specific type of model update. For instance:
> * **Explanation 1**: _“Item B is highly likely to be part of a high-priority order, whereas it was initially considered much less likely.”_ (change-probability)
> * **Explanation 2**: _“The assumption that the task should only be executed when Item B is part of a high-priority order is inaccurate.”_ (delete-rule)
> * **Explanation 3**: _“The loading zone is unlikely to be congested and the task can also be executed if the zone has space available.”_ (add-fact and add-rule)
>
> _Note that the labels in parentheses indicate the corresponding update actions in the model. These were not shown to participants and are included here only for clarification._
>
> Participants are asked to take the perspective of the human operator and compare pairs of explanations (e.g., Explanation 1 vs. Explanation 2). For each pair, they answer the question: "If you were the operator, which explanation would you prefer to receive from Blitzcrank?"
>
> **Evaluation.** To analyze the results, we use the Bradley–Terry model to estimate the relative cognitive cost of each explanation type based on the pairwise preferences collected.
>
> Our  ***computational*** evaluation complements the user study by assessing scalability and runtime on synthetic agent-human model pairs, as described in Section 6.2. Since there is no existing benchmark dataset for this task, we generate the models using a logic-driven process inspired by Example 2 in our paper. We vary the number of facts and rules in the agent model and introduce controlled differences to simulate the kinds of belief divergences a human might have. We will clarify this procedure and include additional examples in the revised version.
>
> **_Q3: The approach is reminiscent of FORTE framework and its probabilistic extension._**
>
> A3: We appreciate the reviewer’s observation. FORTE and its probabilistic extension PFORTE follow the theory revision from examples paradigm. They start with an initial logical model and iteratively fix errors using operations like deleting rules or adding conditions, aiming to improve classification accuracy through small, step-by-step adjustments.
>
> Our framework, by contrast, has a different goal. While FORTE and PFORTE repair models after observing mistakes, we focus on reconciling differences between two probabilistic models: one agent model and one human model. Instead of correcting predictions, we generate minimum-cost explanations that clarify why the agent’s behavior is reasonable. Each model update has a cost, and we use A*-based search to find the explanation that is both logically valid and cognitively aligned.
>
> We will clarify this distinction and cite relevant work in the revised Related Work section.
>
> **_Q4: Scalability and reliance on the current dataset are potential limitations._**
>
> A4: Regarding scalability, our framework supports both exact and approximate explanation generation using A*-based search. The approximate variants leverage enhanced heuristics that significantly reduce runtime while producing suboptimal explanations within theoretical bounds, as demonstrated in our updated results (see A1). These variants remain effective even when exact search becomes infeasible, making the framework robust to larger models.
>
> For the dataset, we note that there is currently no established benchmark for probabilistic model reconciliation. To enable controlled and systematic evaluation, we generate agent–human model pairs (see Example 2 in our paper) using a logic-based process with configurable complexity and model differences. This setup allows us to simulate a wide range of reconciliation scenarios and evaluate performance in a reproducible way. We will clarify this in the revised version.
>
> Finally, the framework is general and supports personalization: if user preferences are known, the cost model can be adapted to produce more user-aligned explanations.

---

> ### Comment · Reviewer_azj8 · 2025-08-05
>
> Dear Authors
>
> Thanks for your detailed reply, I understand the concern on efficiency and scalability,but I agree that more examples would be really helpful in motivating your work.

---

> > ### Author Response · Authors · 2025-08-05
> >
> > Thank you for your suggestion. We agree that more examples would help clarify and motivate the contribution. We will include additional examples in the revised version. For example, consider a simple scenario where a human decides **NOT** to bring an umbrella because they believe the chance of rain is _low_ (e.g., 20\%), based on personal experiences. In contrast, the agent recommends bringing an umbrella because it believes rain is _highly likely_ (e.g., 85\%), based on recent weather data analyses that take into account factors such as cloud coverage, humidity levels, and atmospheric pressure.
> >
> > These differing beliefs under uncertainty can be naturally represented in ProbLog:
> >
> > Human's model $\mathcal{M}_h$:
> > $$\texttt{0.2::rain.}$$
> > $$\texttt{bring-umbrella :- rain.}$$
> >
> > Agent's model $\mathcal{M}_a$:
> > $$\texttt{0.85::rain.}$$
> > $$\texttt{bring-umbrella :- rain.}$$
> >
> > This simple example demonstrates how differences in assumptions lead to divergent decisions, and it motivates the need for model reconciliation to explain and resolve such discrepancies.
> >
> > More realistic and domain-specific examples are already included in our user study (see **A2**), where participants interact with an agent in a warehouse setting. These scenarios involve richer rules and dependencies, better reflecting the complexity of real-world decision-making.
> >
> > In general, any example that involves computing the MPE within a probabilistic logic program (PLP) can be used to illustrate such disagreements. More examples can be found on the official ProbLog website, which we will also reference in the revised version.

---

### Note · Authors · 2025-08-12

We thank the reviewers for their thoughtful feedback. Our aims are threefold: _**formal definition, theoretical analysis, and efficient computation**_ of cost-optimal reconciliation for agent–human ProbLog MPE inconsistencies. Our core contributions are:

 * _**Formal Definition.**_ We present the _first_ formalization of model reconciliation under *uncertainty* for agent–human inconsistencies in *MPE probabilities*, framed as a cost-optimal objective with explicit minimality and correctness.
 * _**Theoretical Analysis.**_  The problem is NP-hard. We justify safe pruning via Theorem 2, show admissibility of the exact heuristics, and offer a greedy method with a $((1+\ln n)$-approximation.
 * _**Computation & Scalability.**_  We design an optimized A* with principled pruning and add tighter admissible and bounded-suboptimal variants that retain guarantees and scale on larger instances.
 * _**Cost Model.**_ We elicit user-aligned costs via Bradley–Terry pairwise comparisons. The costs affect only the final choice of the cost-optimal explanation, not the framework’s formalism, algorithms, or guarantees.

Most reviewers share the following concerns, which we address next.
 * _**Computation & Scalability.**_ The problem is NP-hard. We mitigate this with an optimized A* using principled pruning (Theorem 2) and tighter admissible heuristics, plus greedy/weighted variants with a $(1+\ln n)$-approximation that scale when exact A* times out.
 * _**Motivation.**_ We use PLP as a formal language to encode assumptions, dependencies, and uncertainty, not to simulate human reasoning. PLP supports MPE/MAP and suits reconciliation under formal semantics.
 * _**Generalization.**_ Costs were estimated in one domain, but the study’s role is only to show a practical way to obtain user-aligned cost weights. These costs are plug-in parameters and do not change the formal problem, algorithms, or guarantees. The framework is domain-independent and can use any cost values, regardless of how they are obtained.
 * _**Evaluation of Generated Explanations.**_ We focus on generating explanations for model reconciliation under uncertainty. Since MAP/MPE align with human cognition and are standard in probabilistic reasoning, we reconcile agent–human discrepancies in MPE probabilities under PLP. A broad efficacy study is valuable but _orthogonal_ to our contributions in definition, theory, and computation.

In the revised version, we will incorporate these improvements accordingly.

---

### Decision · Program_Chairs · 2025-09-17

**Decision:**

Accept (poster)

**Comment:**

The work introduces model reconciliation (between an agent and a human) within the realm of probabilistic logic programming. The work's contributions are the formal definition of model reconciliation under uncertainty, theoretical analysis, algorithms for computation of human's reconciled model, and a cost model. The reviewers agree that the submission is well written and organized. Most reviewers expressed concerns about motivation to use PLP for model reconciliation, limited extent of empirical evaluation. Some reviewers also mentioned that the paper would be more appropriate to a general AI conference than to this venue.

The authors provided detailed and constructive feedback with theoretical explanations and empirical evidence during the discussion period, resolving some of the reviewers concerns, however other concerns remained, in particular the motivation to use PLP for model reconciliation and the extent of empirical evaluations. Since all reviewers recommend acceptance,  and the work is an interesting and refreshing view in the modern landscape of agent-human interaction, I second the reviewers' opinion.